# Time-dependent 3D simulations of tropospheric ozone depletion events in the Arctic spring using WRF-Chem

Maximilian Herrmann [1], Holger Sihler [2,3], Udo Frieß [3], Thomas Wagner [2,3], Ulrich Platt [3,4], and Eva Gutheil [1,4]

[1]Interdisciplinary Center for Scientific Computing, Heidelberg University, Heidelberg, Germany
[2]Max-Planck Institute for Chemistry, Mainz, Germany
[3]Institute of Environmental Physics, Heidelberg University, Heidelberg, Germany
[4]Heidelberg Center for the Environment, Heidelberg University, Heidelberg, Germany

**Correspondence:** M. Herrmann (maximilian.herrmann@iwr.uni-heidelberg.de)

**Abstract.** Tropospheric bromine release and ozone depletion events (ODEs) as they commonly occur in the Arctic spring are studied using the open-source software package WRF-Chem. For this purpose, the MOZART-MOSAIC chemical reaction mechanism is extended by bromine and chlorine reactions as well as an emission mechanism for reactive bromine via heterogeneous reactions on ice and snow surfaces. The simulation domain covers an area of 5,040 km x 4,960 km, centered north of Utqiaġvik (formerly Barrow), Alaska, and the time interval from February through May, 2009. Several simulations for different strengths of the bromine emission are conducted and evaluated by comparison with in-situ and ozone-sonde measurements of ozone mixing ratios as well as by comparison with tropospheric BrO vertical column densities (VCDs) from the Global Ozone Monitoring Experiment–2 (GOME-2) satellite instrument. The base bromine emission scheme includes the direct emission of bromine due to bromide oxidation by ozone. Results of simulations with the base emission rate agree well with the observations, however, a simulation with 50% faster emissions performs somewhat better. The bromine emission due to bromide oxidation by ozone is found to be important to provide an initial seed for the bromine explosion. Bromine release due to $N_2O_5$ was found to be important from February to mid March, but irrelevant thereafter. A comparison of modeled BrO with in-situ and MAX-DOAS data hints at missing bromine release and recycling mechanisms on land or near coasts. Consideration of halogen chemistry substantially improves the prediction of the ozone mixing ratio with respect to the observations. Meteorological nudging is essential for a good prediction of ODEs over the three months period.

## 1 Introduction

Ozone is an important constituent of the troposphere due to its high oxidation potential. In the Arctic troposphere, ozone mainly originates from transport and photo-chemical reactions involving nitrogen oxides and volatile organic compounds, resulting in a background mixing ratio of 30 to 50 nmol/mol (ppb). During polar spring, so-called tropospheric ozone depletion events (ODEs) are regularly observed, in which ozone mixing ratios in the boundary layer drop to almost zero levels coinciding with a surge in reactive bromine levels on a time scale of hours to days (e.g. Oltmans, 1981; Bottenheim et al., 1986; Barrie et al., 1988; Hausmann and Platt, 1994; Wagner and Platt, 1998; Frieß et al., 2004; Wagner et al., 2007; Helmig et al., 2012;

Halfacre et al., 2014). ODEs strongly shorten the lifetime of ozone and organic gases, they cause the removal and deposition of mercury as well as the transport of reactive bromine into the free troposphere. During an ODE, ozone is most likely destroyed

by Br atoms in the catalytic reaction cycle (e.g. Barrie et al., 1988; Wang et al., 2019b)

$$Br + O_3 \rightarrow BrO + O_2 \tag{R1}$$

$$BrO + BrO \rightarrow \begin{cases} 2\,Br + O_2 \\ Br_2 + O_2 \end{cases} \tag{R2}$$

$$Br_2 + h\nu \rightarrow 2Br, \tag{R3}$$

resulting in the net reaction

$$2O_3 \rightarrow 3O_2. \tag{R4}$$

The rate-limiting reaction in this cycle is usually the BrO self-reaction (R2) with a reaction rate that is quadratic in the BrO concentration. The source of the reactive bromine is thought to be sea salt, i.e. aerosol, which deposits on the snow (Fan and

Jacob, 1992; McConnell et al., 1992; Platt and Janssen, 1995; Pratt et al., 2013; Simpson et al., 2015; Custard et al., 2017). However, it is not fully understood how the salt bromide is oxidized and how the reactive bromine is released into the air. The most widely accepted emission mechanism is autocatalytic and termed "bromine explosion" (Platt and Janssen, 1995; Platt and Lehrer, 1997; Wennberg, 1999), which consists of the reactions (R1), (R3), and the following two reactions (R5) and (R6)

$$BrO + HO_2 \rightarrow HOBr + O_2 \tag{R5}$$

$$HOBr(g) + H^+(aq) + Br^-(aq) \rightarrow Br_2(g) + H_2O(l). \tag{R6}$$

Reaction (R6) is a heterogeneous reaction, i.e. a reaction involving gaseous components (HOBr) and liquid phase components ($H^+$ and $Br^-$). The concentration of atomic gas-phase bromine doubles in each reaction cycle as can be seen in the following net reaction

$$Br(g) + O_3(g) + HO_2(g) + Br^-(aq) + H^+(aq) \rightarrow 2Br(g) + 2O_2(g) + H_2O(l). \tag{R7}$$

Since $H^+$ ions are consumed, it implies the need for acidic solutions for this reaction to occur, and a pH of at most 6.5 is suggested by Fickert et al. (1999) for this reaction to efficiently occur. A pH-dependence of the $Br_2$ production was shown by Pratt et al. (2013) through field-based experiments.

Other pathways to activate bromide were suggested, involving nitrogen oxides

50   $BrO + NO_2 + M \rightarrow BrONO_2 + M$                    (R8)

$BrONO_2(g) + Br^-(aq) \rightarrow Br_2(g) + NO_3^-(aq),$                   (R9)

as well as a direct emission due to bromide oxidation by ozone (e.g. Oum et al., 1998; Artiglia et al., 2017), which are likely to need sunlight to efficiently occur (Pratt et al., 2013)

55   $O_3(g) + 2Br^-(aq) + 2H^+(aq) \rightarrow Br_2(g) + O_2(g) + H_2O(l).$              (R10)

In the following discussion, the term bromine explosion mechanism includes the original reactions (R1), (R3), (R5), and (R6) as well as reaction (R9), which also generates two bromine atoms out of one gas-phase bromine atom and represents an extended bromine explosion mechanism. Reaction (R10) is considered independently of this terminology as bromide oxidation due to ozone.

60   A further $Br_2$ release mechanism initiated by a reaction of the hydroxyl radical OH with bromide inside the surface layer of the snow grains under sunlight was suggested (Sjostedt and Abbatt, 2008; Pratt et al., 2013). Evidence for this mechanism was found in a laboratory study (Halfacre et al., 2019). The release mechanism may be summarized in the net reaction

$OH(aq) + 2Br^-(aq) + H^+(aq) \rightarrow Br_2(g) + H_2O(l).$                (R11)

A consequence of the reduced ozone levels during an ODE is that reactions of reactive bromine with OH or certain organic 65 species producing chemically inert HBr are favored (essentially reactive bromine is returned to the bromide reservoir), e.g.

$Br + CH_2O + O_2 \rightarrow HBr + CO + HO_2.$                  (R12)

HBr then deposits into the ground or onto aerosols, ultimately terminating the ODE. Chlorine and iodine play a smaller role for the occurence of ODEs (Thompson et al., 2015). The reaction of methane with chlorine atoms quickly produces chemically inert HCl. Since Cl-atoms react with $CH_4$ (while Br- and I-atoms do not) and due to the large abundance of methane in the 70 atmosphere, chlorine explosions cannot occur in the atmosphere. The iodine concentration ($I^-$ and $IO_3^-$) is approximately twenty times smaller than bromide in seawater (Luther et al., 1988; Grebel et al., 2010), which is likely the reason why detectable amounts of gaseous iodine were rarely found in the Arctic and the Antarctic (Wittrock et al., 2000; Schönhardt et al., 2008; Saiz-Lopez et al., 2007; Atkinson et al., 2012; Zielcke, 2015; Raso et al., 2017). Both iodine and chlorine, however, still may play a role due to interhalogen reactions

75   $BrO + XO \rightarrow BrX + O_2$                       (R13)

$BrX + h\nu \rightarrow Br + X,$                        (R14)

with X = Cl or I, that occur faster by an order of magnitude (Atkinson et al., 2007) than the BrO self reaction (R2).

Similarly, chloride can speed up bromine activation (Simpson et al., 2007a)

$$HOBr(g) + H^+(aq) + Cl^-(aq) \rightarrow BrCl(aq) + H_2O(l),$$ (R15)

and aqueous BrCl can further be converted into $Br_2$

$$BrCl(aq) + Br^-(aq) \rightarrow Br_2(g) + Cl^-(aq).$$ (R16)

ODEs are observed mostly in the polar spring. During winter, radical bromine chemistry cannot occur due to the lack of sunlight. Temperatures below $-20°C$ are likely to favor the occurrence of ODEs (Tarasick and Bottenheim, 2002; Pöhler et al.,
2010). However, Bottenheim et al. (2009) observed ODEs at $-6°C$ and Halfacre et al. (2014) found no apparent temperature dependence for the presence of an ODE in ozone measurements at five buoys across the Arctic. Shallow boundary layers are also likely to be beneficial (Wagner et al., 2001; Frieß et al., 2004; Lehrer et al., 2004; Koo et al., 2012), since they increase the speed of the auto-catalytic bromine release by confining the released bromine to a smaller space. The age of the sea ice is also an important factor. Snow covering First-year (FY) ice, which has more accessible salt than multi-year (MY) ice, is expected
to be the main source of bromine (Simpson et al., 2007b; Abbatt et al., 2012). Despite being often depleted in bromide, snow covering MY ice may still play an active role in the release of reactive bromine (Peterson et al., 2019). Pratt et al. (2013) did not directly observe $Br_2$ emissions from the sea ice, which is likely due to a higher pH of the sea ice due to buffering (Wren and Donaldson, 2012). ODEs are much less pronounced in polar fall with rare measurements of partial ODEs in the Antarctic (Nasse, 2019), because most of the brine covering FY ice will have drained away during the summer melt (Simpson et al.,
2007b) even though meteorological conditions are similar to those in spring time.

Snow covering land surfaces may also play an active role in the release of $Br_2$, as several studies suggest (Simpson et al., 2005; Peterson et al., 2018). Custard et al. (2017) simultaneously measured $Br_2$, BrCl, and $Cl_2$ in the snowpack interstitial air and also provided estimates of $Br_2$ and $Cl_2$ emission rates. McNamara et al. (2020) measured the release of BrCl from snow surfaces and the dominant pathways of BrCl were identified in a box model simulation. Thomas et al. (2011) extended the 1D
model MISTRA with a snow pack module and validated their results with observations at Summit, Greenland. They found the solar actinic flux to be the main driver of reactive bromine release from the liquid-like layer (LLL) of the snow grain surface and a dependence of bromine release from the LLL on the OH concentration in the LLL. Wang and Pratt (2017) attributed approximately 20% of the total $Br_2$ production to the mechanism of snow $Br_2$ production. Wang et al. (2019b) measured atomic bromine and related it to BrO and snow-released $Br_2$, finding three to ten times higher levels of atomic bromine than previous
estimates suggested.

From the outline above it is clear that ODEs are a complex function of chemistry and meteorology, therefore 3D simulations are useful to learn about the interaction of meteorology and chemistry in generating ODEs. Earlier studies estimated boundary layer BrO from measurements of satellite BrO vertical column densities (VCDs) (e.g. Wagner and Platt, 1998; Zhao et al., 2008) by estimating the BrO release from sea-salt aerosols produced from abraded frost flowers (Kaleschke et al., 2004; Zhao
et al., 2008) or from blowing snow events (Yang et al., 2008, 2010). Toyota et al. (2011) reproduced major features of satellite

BrO VCDs and in-situ measurements using a simple parameterization of bromine emissions from bulk ice and snow with the 3D air quality model Global Environmental Multiscale model with Air Quality processes (GEM-AQ). Falk and Sinnhuber (2018) integrated this mechanism into the ECHAM/MESSy Atmospheric Chemistry (EMAC) model, investigating and reproducing important features of ODEs for a full annual cycle.

In the present study, the regional 3D online numerical weather prediction system WRF-Chem is used to investigate the ODEs during Arctic spring from February 1 through May 1, 2009 since for this period of time, extensive data from observations are available from the NOAA institute or collected as part of the Ocean-Atmosphere Sea-Ice Snowpack (OASIS) field initiative for comparison with the numerical results. The chemical reaction scheme MOZART-MOSAIC is extended by bromine and chlorine reactions to study their impact on the ODEs. The emission scheme developed by Toyota et al. (2011) is adopted and a

parameter study for the reactive surface ratio (Cao et al., 2014) of the ice/snow surface is performed.

## 2   Model

First, the configuration of WRF-Chem (Grell et al., 2005; Skamarock et al., 2008) will be presented, then the modifications to the standard configuration will be discussed and the initial and boundary conditions will be provided.

### 2.1   Configuration of WRF-Chem

The physical area (displayed in Fig. 1) of 5,040 km x 4,960 km, centered north of Utqiaġvik, is modeled for the time interval of February 1, 2009 through May 1, 2009, for which GOME-2 data with a stratospheric correction for BrO VCDs (Sihler et al., 2012) as well as surface ozone and ozone sonde data are available for model evaluation.

    The software Weather Research and Forecasting model coupled with Chemistry (WRF-Chem) version 3.9 is employed. WRF-Chem (Skamarock et al., 2008; Grell et al., 2005) is a state-of-the-art regional numerical weather prediction system

with online computation of chemistry. Table 1 summarizes the configuration of the software. The physics modules are chosen following recommendations of the Polar WRF community (Bromwich et al., 2009; Wilson et al., 2011; Bromwich et al., 2013), the modules include the meteorology and the emission, transport, mixing, chemical reactions of trace gases as well as aerosols.

    The simulation domain is centered north of Utqiaġvik using the polar stereographic projection at a true latitude of $83°$ with a reference longitude of $156°$ W. A horizontal grid resolution of 20 km for the 5,040 km x 4,960 km domain is employed,

allowing comparison to GOME-2 BrO satellite data (Sihler et al., 2012) with a resolution of approximately 40 km x 30 km. In vertical direction, 64 non-equidistant grid cells with a finer resolution near the ground are used, starting with approximately 25 m at the ground level. Half of the grid cells used in the present study are in the first 2 km of the atmosphere, allowing a detailed representation of the Arctic boundary layer. The vertical grid is provided in the supplement of this manuscript.

    The meteorological time step of one minute is chosen to fulfill the Courant criterion. Chemistry is updated between every

meteorology time step, and radiative transfer is updated every tenth meteorological time step.

    In the present model, the Mellor-Yamada-Janjic (MYJ) PBL scheme (Mellor and Yamada, 1982; Janjić, 1990) is employed, which is a 1.5-order local turbulence closure model. Prognostically determined turbulent kinetic energy is used to determine the

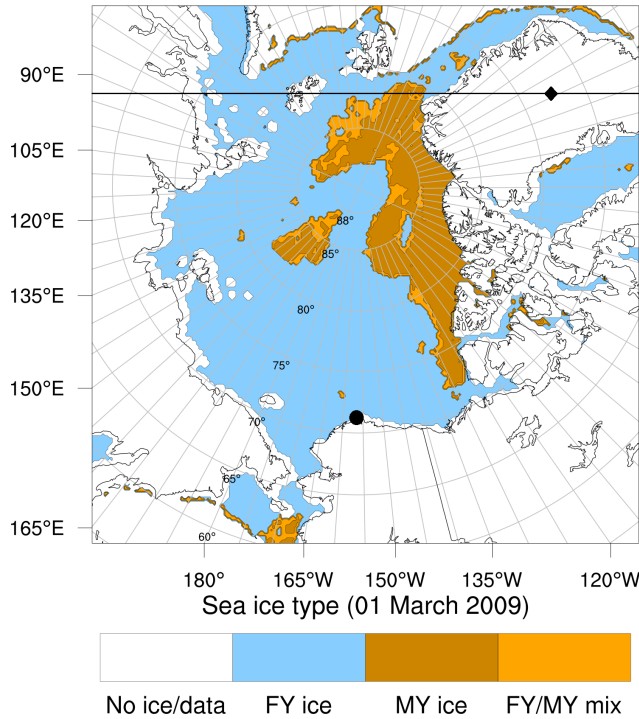

**Figure 1.** Domain of the simulations presented in this publication and sea ice type (Aaboe et al., 2017) with the locations of (●) Utqiaġvik, Alaska, and (♦) Summit, Greenland, respectively. For latitudes larger than 88°, missing sea ice type data is filled up with FY ice. The horizontal line refers to the $x$-coordinate in Fig. 6.

eddy diffusion coefficients. The MYJ PBL scheme is best suited for stable to slightly unstable conditions (Mellor and Yamada, 1982).

**2.2 Gas-phase chemistry**

WRF-Chem offers several implementations of chemical reaction schemes. In the present study, the MOZART-MOSAIC mechanism based on MOZART-4 (Model for Ozone and Related chemical Tracers) gas-phase chemistry (Emmons et al., 2010a) is used which includes 85 gas-phase species, 237 gas-phase reactions, 49 photolysis reactions. Additional 18 gas-phase species, 73 gas-phase reactions, and 13 photolysis reactions (Herrmann et al., 2019) account for the bromine and chlorine chemistry

(termed "full chemistry", see Tab. 2). Observations of reactive iodine in the arctic region (Zielcke, 2015; Raso et al., 2017) suggest only low mixing ratios of iodine. Even though small mixing ratios of iodine can significantly enhance ozone depletion (Raso et al., 2017), iodine is neglected due to the uncertainties in the abundance of iodine in the arctic atmosphere and in snowpacks. The photolysis rates are calculated with the "Updated TUV" scheme (Madronich et al., 2002), which already contains the halogen photolysis reaction rates. The added bromine and chlorine chemical reactions are provided in the supplement.

**Table 1.** Summary of the configuration of WRF-Chem.

| Parameter | Setting |
| --- | --- |
| Longwave radiation | LW RRTMG scheme (Iacono et al., 2008) |
| Shortwave radiation | SW RRTMG scheme (Iacono et al., 2008) |
| Microphysics | WSM 6-class graupel scheme (Hong and Lim, 2006) |
| Land-surface model | Noah Land-Surface Model (Niu et al., 2011) |
| Surface-layer model | Monin-Obukhov (Janjic Eta) Similarity scheme (Janjić, 1996) |
| Boundary-layer model | Mellor-Yamada-Janjic (MYJ) scheme (Mellor and Yamada, 1982) |
| Cumulus parameterization | Grell 3D ensemble scheme (Grell, 1993) |
| Initial and boundary data | ERA-Interim (Dee et al., 2011), MOZART-4 (Emmons et al., 2010a) |
| Sea ice data | OSI-403-c (Aaboe et al., 2017) |
| Sea surface temperature data | RTG_SST high resolution (Thiébaux et al., 2003) |
| Time step | 1 min |
| Simulated time range | February 1, 2009 – May 1, 2009 |
| Nudging | included, see text |
| Horizontal resolution | 20 km |
| Longitude and latitude | 252 × 248 horizontal grid cells |
| Vertical grid size | 64 eta levels |
| Vertical size of the first cell | $\approx 25$ m |
| Pressure at top boundary | 50 hPa |
| Chemistry mechanism | MOZART-MOSAIC (Emmons et al., 2010b) |
| | plus bromine and chlorine reactions (see supplement) |
| Aerosols | MOSAIC 4 bin aerosols (Zaveri et al., 2008) |
| Photolysis scheme | Updated TUV (Madronich et al., 2002) |
| Emissions | EDGAR-HTAP (Janssens-Maenhout et al., 2012) |
| Bioemissions | MEGAN (Guenther et al., 2006) |

**2.3 Aerosol-phase chemistry**

The MOZART-MOSAIC mechanism employs four-bin MOSAIC aerosols (Zaveri et al., 2008). In WRF-Chem, MOSAIC is implemented using a sectional approach, where size bins are defined by the upper and lower dry particle diameters. In MOSAIC, mass and number density for each bin are considered and the processes of nucleation, coagulation, condensation, evaporation, and aerosol chemistry are modeled. The mass transfer rate $k_{i,m}$ for gas species $i$ and aerosol size section $m$ is

calculated using the parameterization (Wexler and Seinfeld, 1991)

$$k_{i,m} = 4\pi R_{\mathrm{p},m} D_{\mathrm{g},i} N_m f(\mathrm{Kn}_m, \gamma_i), \tag{1}$$

where $D_{g,i}$ is the gas diffusivity of species $i$, $R_{p,m}$ is the wet mean particle radius of size bin $m$, $N_m$ the number density of size bin $m$, and $Kn_m = \lambda/R_{p,m}$ is the Knudsen Number of size bin $m$ with the free mean path $\lambda$. $f(Kn_m, \gamma_i)$ is the transition regime correction factor (Fuchs and Sutugin, 1971) and accounts for the interfacial mass transport limitation

$$f(Kn_m, \gamma_i) = \frac{0.75\gamma_i(1+Kn_m)}{Kn_m(1+Kn_m)+0.283\,\gamma_i\,Kn_m+0.75\gamma_i}, \tag{2}$$

where $\gamma_i$ is the accommodation coefficient for gas-phase species $i$ taken from the CAABA/MECCA model (Sander et al., 2011). Aerosol forms of bromine are currently not implemented in the MOSAIC framework and are treated as gas-phase species. The transfer reactions of bromine gas-phase species X to aerosol-size bin $m$ are assumed to produce species $X_{aq,m}$ as

$$HBr(g) \rightarrow HBr_m(aq) \tag{R17}$$

$$HOBr(g) \rightarrow HOBr_m(aq) \tag{R18}$$

$$BrONO_2(g) \rightarrow HOBr_m(aq) + HNO_3(g), \tag{R19}$$

which may produce gas-phase $Br_2$ (McConnell et al., 1992; Peterson et al., 2017)

$$HOBr_m(aq) + HBr_m(aq) \rightarrow Br_2(g). \tag{R20}$$

Reactions (R17)-(R20) may only occur if the aerosol is in a liquid state, and in addition, reaction (R20) requires the aerosol to have a pH of 6 or less. The heterogeneous reactions and parameters required to calculate the reaction rates are listed in the supplement. Heterogeneous BrCl production (reactions (R15) and (R16)) is not implemented in the model.

## 2.4 Bromine emission scheme

Emissions of bromine species on snow surfaces are parameterized following Toyota et al. (2011). Numerically, bromine emissions are coupled to vertical diffusion. In WRF-Chem, vertical (turbulent) diffusion for each species and horizontal grid cell is solved using a Peaceman-Rachford Alternating direction implicit method (Peaceman and Rachford, 1955). The bromine emissions are added as boundary conditions to the tridiagonal diffusion matrix. For the surface emission in reactions (R6), (R9), and (R10), the boundary flux for instance of (R6), $F_d(Br_2|HOBr)$ for $Br_2$ due to HOBr is

$$F_d(Br_2|HOBr) = \beta\rho_{d,0}v_d(HOBr)\,[HOBr]_0 \tag{3}$$

where $\rho_{d,0}$ is the dry air density of the lowest grid cell and $[HOBr]_0$ is the HOBr mixing ratio in the lowest grid cell. The species-dependent deposition velocity $v_d \approx 1\ cm\ s^{-1}$ is calculated using the WRF-CHEM Wesely deposition module (Wesely, 1989) under an additional assumption of near-zero surface resistance. Thus, the turbulent transfer resistance dominates the

deposition velocity, and the bromine emissions increase with larger wind speeds. $\beta \geq 1.0$ is the reactive surface ratio (Cao et al., 2014) of the ice/snow surface, accounting for non-flat surfaces such as ice/snow and frost flowers. For simplicity, $\beta$ is set as a global value in this study, allowing to investigate the strength of bromine emissions in a parameter study. For the direct emission of bromine due to ozone oxidation of bromide, see reaction (R10) above, the factor $\alpha$ is used to control the emission probability

$$O_3 \longrightarrow \alpha Br_2 \tag{4}$$

and

$$F_d(Br_2|O_3) = \alpha \beta \rho_{d,0} v_d(O_3) [O_3]_0. \tag{5}$$

The value of $\alpha$ is parameterized with a dependence on the solar zenith angle SZA (Toyota et al., 2011)

$$\alpha(SZA) = \begin{cases} 0.1\% & \text{if SZA} > 85° \\ 7.5\% & \text{otherwise} \end{cases} \tag{6}$$

The deposition velocity for ozone is dominated by the surface resistance (Wesely, 1989), leading to $v_d(O_3) \approx 0.01 \text{ cm s}^{-1}$. An emission mechanism relating to the bromide oxidation by the hydroxyl radical, see reaction (R11), is currently not implemented in the model. All sea ice is assumed to be snow covered for the simulated time range. On snow covering FY ice, it is assumed that the bromide content is infinite, so that unrestricted gaseous bromine emissions are possible, and emissions of $Br_2$ due to $O_3$ and $N_2O_5$ depositions are only active on snow covering FY ice. On snow covering MY ice, no bromide content but infinite chlorine is assumed. HOBr depositions only release $Br_2$ up to the combined depositions of gaseous and aerosol HBr whereas excess HOBr depositions release BrCl. On snow-covered land, neither bromide nor chloride content is assumed, so that excess HOBr depositions are lost. A list of the depositions and emissions added to the MOZART mechanism can be found in the supplement. Sunlit condensed-phase $Br_2$ production without any depositions of gas-phase species (Pratt et al., 2013; Wang et al., 2019b) and oceanic emissions of very short-lived brominated species are currently not considered in the model.

## 2.5 Initial and boundary conditions

ERA-Interim (Dee et al., 2011) is used to generate both the initial and boundary meteorological and sea ice cover data. The ERA-Interim Reanalysis was found to perform well in polar regions in various studies (e.g. Bracegirdle and Marshall, 15 Oct. 2012; Bromwich et al., 2016) (e.g. Bracegirdle et al, 2012, Bromwich et al, 2016) and was successfully used in various modeling studies in polar regions (e.g. Hines et al., 01 Jun. 2015; Cai et al., 2018), which is why it was chosen in the present study. Nudging of temperature, horizontal wind speed, humidity, and surface fields to ERA-Interim data ensures the validity of the simulation meteorology over the simulated three month period. The idea of the present work is not to try to make meteorological predictions (which would not be meaningful anyway on the timescale of a few months) but rather to model chemistry under meteorological conditions prevailing over a particular period of time. Nudging is active for the entire duration of the simulation and is inactive inside the boundary layer. The nudging timescale is set to one hour. MOZART-4 results driven

by GEOS-5 meteorological fields are used as initial and boundary data for all non-halogen species (Emmons et al., 2010a). For
most halogen species, initial and boundary conditions are set to near-zero values. The initial mixing ratio of HBr and $Br_2$ are
set to 0.3 ppt in the lowest 200 m of the atmosphere. The mixing ratio of $CHBr_3$ is fixed to 3.5 ppt (Toyota et al., 2014). The
bromide oxidation of ozone in the dark for an ozone deposition velocity of 0.01 cm s$^{-1}$, a boundary layer height of 200 m,
an emission probability of $\Phi = 0.001$, and 40 nmol mol$^{-1}$ ozone will release approximately 2 pmol mol$^{-1}$ $Br_2$ on FY ice
per day. This emission rate is assumed to prevail for all simulations with active halogen chemistry. The chosen initial halogen
concentrations and the fixed mixing ratio of $CHBr_3$ thus are irrelevant. The RTG_SST high-resolution dataset (Thiébaux et al.,
2003) is used for the sea surface temperature (SST). In the present model, it is differentiated between FY and MY sea ice in
order to estimate bromine emissions. For this purpose, the OSI-403-c sea ice type dataset (Aaboe et al., 2017) is used. The
original dataset does not provide values for latitudes larger than about 88° due to a lack of satellite measurements for these
latitudes. In the present study, these values are filled with first year sea ice. Figure 1 shows the simulation domain and the
locations of FY and MY sea ice. Grid cells with a mixed FY/MY sea ice type are treated as multi-year sea ice in the bromine
emission mechanism described above. Sea ice cover, SST, and sea ice type are updated online during the numerical simulations.
EDGAR-HTAP (Janssens-Maenhout et al., 2012) and MEGAN (Guenther et al., 2006) are used as antropogenic emissions and
bioemissions, respectively.

## 2.6 Conducted simulations & observations for comparison

The conducted simulations are summarized in Tab. 2. Five different observational data sets are used for comparison to the
simulation results:

- ground-based in-situ ozone measurements at Utqiaġvik, Alaska, and Summit, Greenland (McClure-Begley et al., 2014).

- ground-based in-situ BrO measurements at Utqiaġvik, Alaska (Liao et al., 2012).

- vertical profiles of the ozone mixing ratio derived from ozone-sonde measurements at Utqiaġvik (Oltmans et al., 2012).

- vertical profiles of the BrO mixing ratio derived from MAX-DOAS measurements at Utqiaġvik (Frieß et al., 2011).

- maps of vertical BrO column densities from GOME-2 satellite measurements (Sihler et al., 2012).

For comparison of the observations and the simulations, three different statistical parameters are used. For model variable $M$
and the corresponding observation variable $O$, the Pearson correlation R, the mean bias MB, and the root mean square error
RMSE are calculated by

$$\text{R} = \frac{\langle (M - \langle M \rangle)(O - \langle O \rangle) \rangle}{\sigma_M \sigma_O} \tag{7}$$

$$\text{MB} = \langle M - O \rangle \tag{8}$$

$$\text{RMSE} = \sqrt{\langle (M - O)^2 \rangle}, \tag{9}$$

where $\langle \ \rangle$ is the mean and $\sigma_M$ and $\sigma_O$ denote the standard deviations of $M$ and $O$, respectively.

**Table 2.** Parameter variation in the simulations.

| condition | reactive surface ratio $\beta$ | meteorological nudging | time period | chemistry |
|:---:|:---:|:---:|:---|:---|
| 1 | 0.0 | on | Feb. 1, 2009 – May 1, 2009 | no halogen chemistry |
| 2 | 1.0 | on | Feb. 1, 2009 – May 1, 2009 | full |
| 3 | 1.5 | on | Feb. 1, 2009 – May 1, 2009 | full |
| 4 | 2.0 | on | Feb. 1, 2009 – May 1, 2009 | full, $\alpha = \mathrm{const} = 0.001$, cf. Eq. (6) |
| 5 | 1.5 | on | March 16, 2009 – May 1, 2009 | full |
| 6 | 1.5 | off | Feb. 1, 2009 – May 1, 2009 | full |

### 2.6.1  Retrieval of the tropospheric BrO VCD from GOME-2 observations

The tropospheric BrO vertical column density (VCD) is derived from GOME-2 observations as described in detail by Sihler et al. (2012). GOME-2 is a UV/visible/near-IR spectrometer with moderate spectral resolution aboard the MetOp-A satellite (Callies et al., 2000; Munro et al., 2006, e.g.) which was launched in 2006. With a swath-width of 1,920 km, almost global coverage is achieved every day. In polar regions, the same location is observed several times during one day. The ground pixel size is approximately 80 km × 40 km.

The atmospheric BrO absorption is analyzed in the spectral range from 336-360 nm. In order to obtain the tropospheric BrO column, the stratospheric BrO column is estimated using the simultaneously retrieved stratospheric columns of $O_3$ and $NO_2$. In the final step, the retrieved tropospheric BrO SCD is converted into the tropospheric BrO VCD using simultaneous measurements of $O_4$ and the radiance at 372 nm. Finally, the retrieved BrO VCDs are filtered and only measurements above a chosen sensitivity threshold of 0.5 for the AMF of the lowest 500 m are used. More details on the data analysis are provided
by Sihler et al. (2012)

### 2.6.2  Retrieval of BrO vertical profiles from MAX-DOAS

Vertical profiles of BrO are derived from Multi-Axis Differential Optical Absorption Spectroscopy (MAX-DOAS) measurements during the OASIS campaign at Utqiaġvik between February to April 2009 as described by Frieß et al. (2011). In brief, BrO and aerosol profiles are retrieved on a vertical layers of 100 m thickness in the lowermost 2 km of the atmosphere with
a temporal resolution of 15 min using the HEIPRO algorithm (Frieß et al., 2019). HEIPRO is based on the well-established optimal estimation method (Rodgers, 2000), with slant column densities (SCDs) of atmospheric trace gases observed at different elevation angles serving as measurement vector. In a first step, aerosol extinction vertical profiles are determined using the observed optical thickness of the oxygen collision complex $O_4$ as a proxy for the atmospheric light path (Frieß et al., 2006). In a second step, BrO vertical profiles are retrieved using BrO slant column densities, together with the aerosol extinction profiles
retrieved in the first step. The limited information content of MAX-DOAS measurements requires the usage of appropriate a priori aerosol and BrO vertical profiles as described in Frieß et al. (2011). Averaging kernels $A = \frac{\partial \hat{x}}{\partial x}$ quantify the sensitivity of the retrieved profile $\hat{x}$ to the true profile $x$. In order to account for the limited vertical resolution of MAX-DOAS measure-

ments and to allow for a quantitative comparison of model and measurement, modeled vertical profiles are convoluted with the MAX-DOAS averaging kernels according to Rodgers and Connor (2003):

$$\tilde{x}_m = x_a + A(x_m - x_a) \tag{10}$$

Here, $x_m$ is the modelled and $x_a$ the a priori BrO profile. It is important to note that the vertical sensitivity strongly depends on visibility that varied strongly during the OASIS campaign due to frequent storms with blowing snow.

## 3  Results and discussion

In the following, the results of the six different simulations are compared to the measurements described in section 2.6.

### 3.1  Surface ozone and meteorology at Utqiaġvik and at Summit

The NOAA and ESRL Global Monitoring Division Surface Ozone (McClure-Begley et al., 2014) measurements near Utqiaġvik and Summit are compared to the simulation results for the numerical grid cell closest to the observation site under consideration where the numerical results in the lowest grid cell are used. The temperature at 2 m, wind speed, and wind directions at 10 m of the Barrow Atmospheric Baseline Observatory (Mefford et al., 1994) are compared to the corresponding simulated surface fields.

Figure 2 shows simulated and observed temperatures, $T$ in 2 m height and wind speeds $u$ in 10 m height at Utqiaġvik. Simulations 1-5 share the meteorology shown in the left of Fig. 2 whereas results of simulation 6 with deactivated meteorological nudging are shown in the right of Fig. 2. The first eleven days in February are very cold, reaching temperatures as low as -40°C and the wind speed is very low during this period of time, which is likely to inhibit BrO emission due to the wind dependence of the emission. Both the wind speed and the temperature increase during the following three weeks, wind

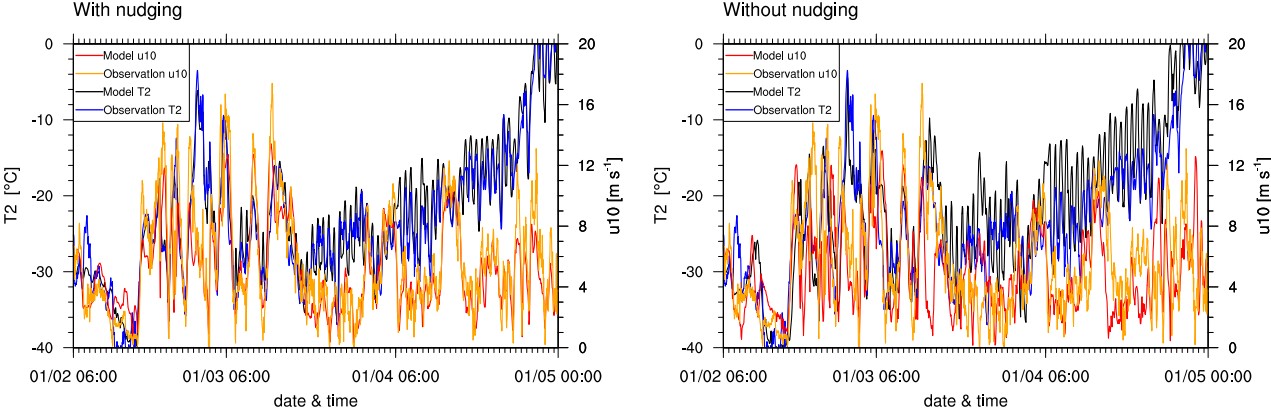

**Figure 2.** 2 m temperature and 10 m wind speed at Utqiaġvik in February through May 2009. Measurements are taken from the Barrow Atmospheric Baseline Observatory (Mefford et al., 1994).

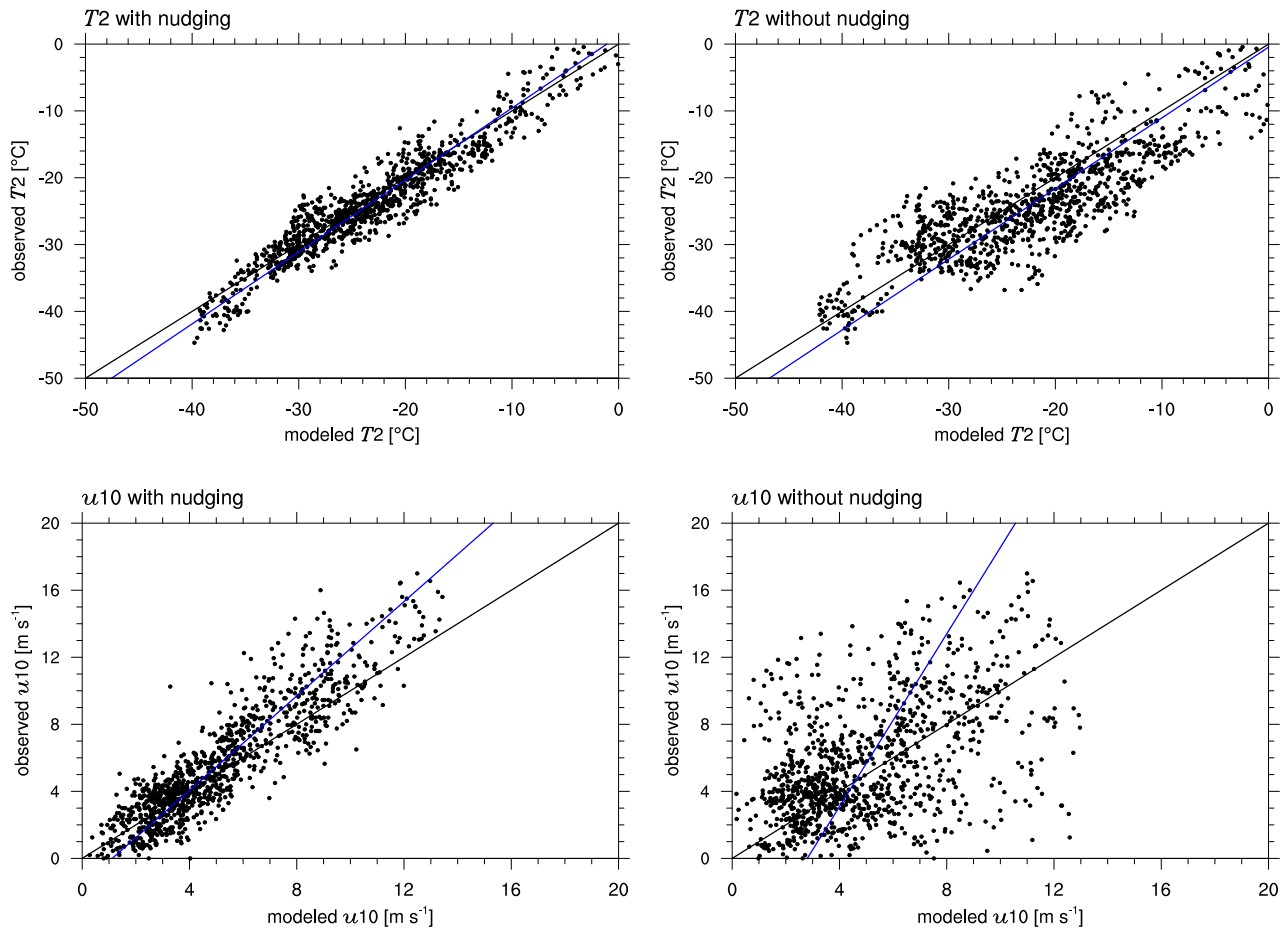

**Figure 3.** Correlation of observed and modeled temperature and wind speed at Utqiaġvik for the complete time range from February 1, 2009 through May 1, 2009. The black and blue lines show perfect agreement and the regression line of the simulation and the observation, respectively.

speeds increase to values up to 16 m s$^{-1}$ and temperature reaches up to -5°C. On February 21, 23, and March 1, wind speed is notably under-predicted by the model with nudging. Both temperature and wind speed vary strongly during that time. From mid March onwards, temperature increases gradually with fewer day to day variations compared to the previous weeks. Simulations 1-5 predict both temperature and wind speed very well during this time period with the exception of under-predictions of wind speed occurring on March 16-17 and in the end of April. Simulation 6 produces higher errors in the second half of the simulation where temperature is consistently too large by several degrees in April and over-predictions of wind speed on March 18-22, April 22 and April 29. The results of simulation 6 appear not to be very realistic.

Figure 3 shows the correlation of the observed (vertical axis) and the modeled (horizontal axis) temperatures, where a correlation of unity applies if the data lie on the diagonal marked in the figure. Shown in blue is the regression line, for which

**Table 3.** Meteorology statistics at Utqiaġvik.

| variable | condition | R | average of the simulated variable | MB | RMSE |
|---|---|---|---|---|---|
| 2m temperature | 1-5 | 0.962 | -22.7°C | 0.547°C | 2.51°C |
| 2m temperature | 6 | 0.874 | -21.5°C | 1.71°C | 5.05°C |
| 10m wind speed | 1-5 | 0.903 | 5.13 m s$^{-1}$ | -0.518 m s$^{-1}$ | 1.64 m s$^{-1}$ |
| 10m wind speed | 6 | 0.492 | 4.99 m s$^{-1}$ | -0.655 m s$^{-1}$ | 3.28 m s$^{-1}$ |
| 10m wind direction | 1-5 | 0.801 | 131° | 3.76° | 55.4° |
| 10m wind direction | 6 | 0.423 | 157° | 29.51° | 100.8° |

the observed and measured variables are assumed to be the independent and dependent variables, respectively. The results of the entire simulation period are displayed, where the first week should be considered as spin-up period. For simulations 1-5, there is an overestimation of the temperature when it is cold, which is likely due to the lowest temperatures occurring during the spin-up time during which the modeling errors are larger compared to other times. The ERA-Interim Reanalysis is known to have a warm bias for temperatures below -25°C (Wang et al., 2019a), which may also explain the deviations. Simulations 1-5

perform well throughout the simulation in contrast to simulation 6 with no nudging. In simulations 1-5, a maximum deviation in temperature of about 8°C occurs and in simulation 6, a stronger temperature difference of up to 20°C is observed.

The statistical parameters, cf. Eq. (9), at Utqiaġvik for the entire time range are shown in Table 3. The simulations with nudging perform better in all regards, emphasizing the necessity of data assimilation. Temperature is predicted best with almost perfect correlation and relatively small mean bias and RMSE. Temperature is over-predicted in all simulations by approximately

0.55°C and 1.71°C for simulations 1-5 and 6, respectively. Colder temperatures are generally favorable for ODEs, both by changing the boundary layer configuration and affecting chemical reaction constants, which could result in an underestimation of ODEs. Both wind speed and direction are predicted less accurately, which might result in wrong source locations or times of the occurrence of ODEs; this is likely to explain some of the differences between simulations and observations. Wind speed is underestimated on average by about 0.52 m s$^{-1}$ and 0.66 m s$^{-1}$ for simulations 1-5 and 6, respectively, which may contribute

to a slight underestimation of bromine emissions due to the dependence of the deposition velocity on wind speed. The Barrow Meteorological Station (BMET) Handbook (Ritsche, 2004) mentions an instrument accuracy of 0.17 m s$^{-1}$ for wind speeds between 0.4 and 75 m s$^{-1}$, a 5.6° wind direction resolution, and 0.25°C instrument accuracy for temperatures between -65 to -20°C. The RMSE is at least one magnitude higher than the mentioned instrument accuracies and resolutions for all simulations, so that the errors of the observations can be neglected in comparison to the model errors.

Figure 4 shows modeled and observed surface ozone and BrO at Utqiaġvik and at Summit. Only results of simulations 1 and 3 are shown for visual clarity. Figure S1 of the supplement displays ozone mixing ratios modeled by simulations 1-4 and 6. The correlations of modeled and observed ozone can be seen in Figure 5. Statistics are summarized in Table 4. Simulation 2-5 perform considerably better than simulation 1 for which halogen chemistry is turned off. Simulation 3 with enhanced emission performs best with the correlation increasing from -0.31 to 0.644 compared to simulation 1. Quite a few ODEs

are not captured by simulation 4, for which the emission probability for bromine emissions due to ozone under sunlight are

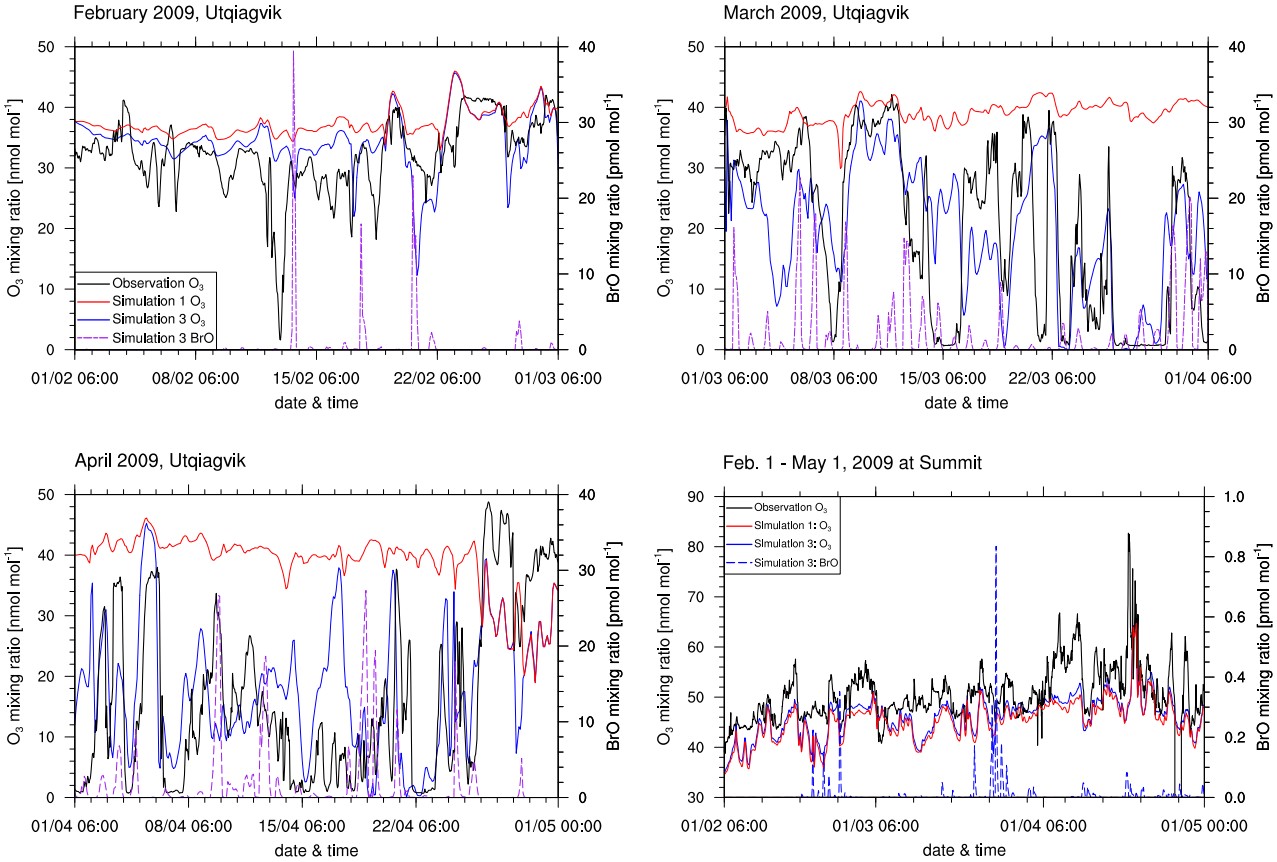

**Figure 4.** Ozone at Utqiaġvik and at Summit from observations and simulations 1 (no halogens) and 3 (increased emissions, $\beta = 1.5$). Modeled BrO mixing ratios are also shown. The figures share the legend.

reduced from 7.5% to 0.1%. Thus, direct emissions of bromine due to ozone are nearly completely turned off in simulation 4. This suggests a strong underestimation of bromine emissions without a direct emission of bromine due to ozone. A possible conclusion is that the bromine explosion mechanism is insufficient to explain ODEs in the Arctic, or the present bromine explosion scheme is incomplete for instance with respect to emissions of bromide containing aerosols due to blowing snow and/or regions of increased $\beta$ such as frost flowers. On March 4 a ODE is predicted by simulation 4 which, however, is not seen in the observations. The model predicts too large wind speeds for the preceding days, causing larger BrO emissions that ultimately result in a predicted ODE being advected to Utqiaġvik. For the first three weeks of February, the observations and results of simulations 2-6 are similar to these of simulation 1 in which halogen reactions are turned off, but afterwards, they differ increasingly. This suggests a weak initial influence of halogen chemistry during the first three weeks of February which might be due to the low wind speeds during this time or due to the weak solar irradiation. Partial ODEs occur on February 14, 17, 19, and 22, 2009. The first full ODE in the observations occurred on the February 13, which is predicted by the model only as a partial ODE with one day of delay. The partial ODE observed on February 17 is found in simulations 2-5 with a delay of

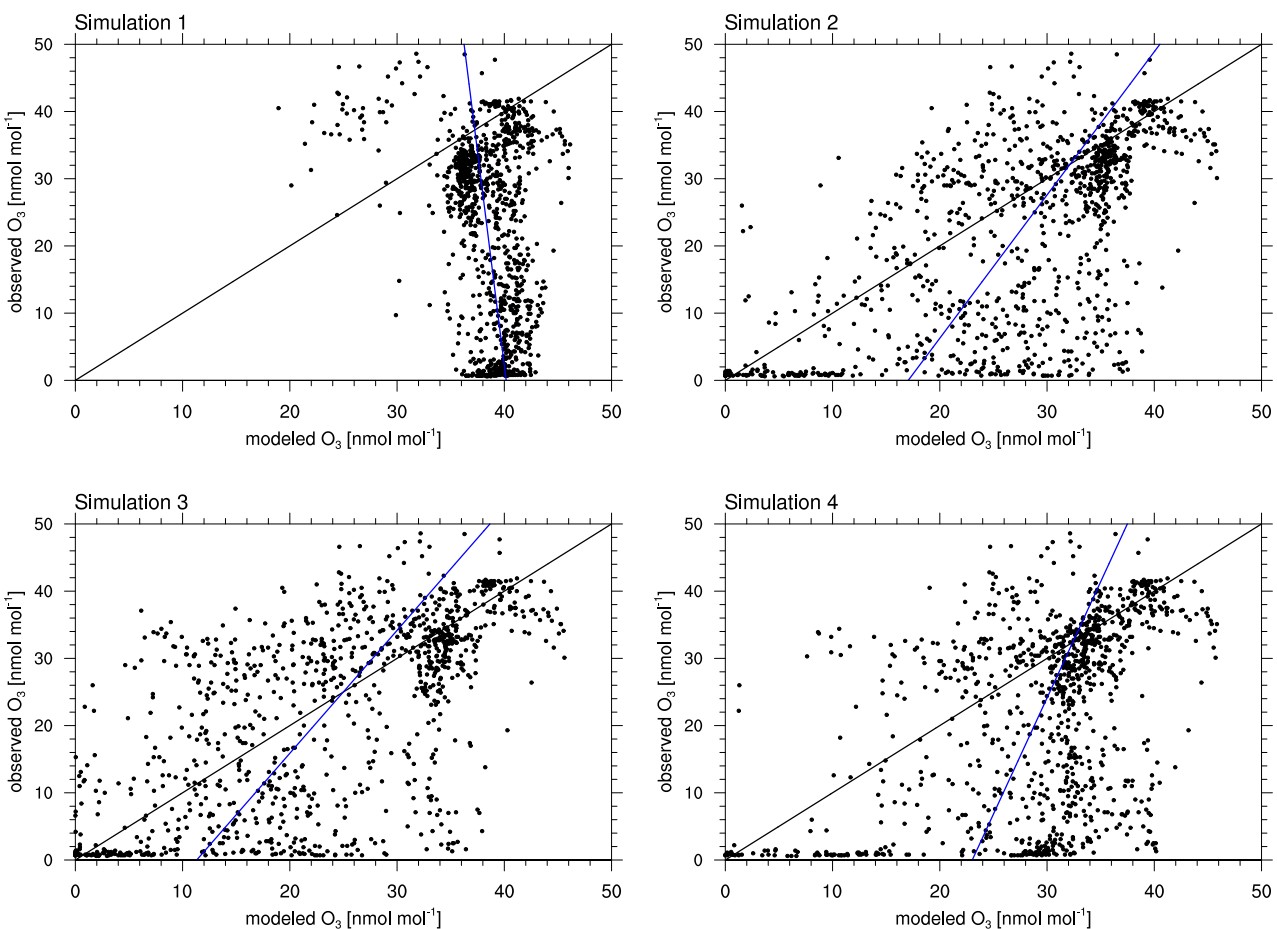

**Figure 5.** Correlation of observed and modeled ozone at Utqiaġvik for the complete time range of February 1, 2009 through May 1, 2009. The black and blue lines show perfect agreement and the regression line of simulation and observation, respectively.

a few hours; simulations 3 and 4 find a stronger ozone depletion more consistent with the observations. On February 21, 2009, simulations 2-3 and simulations 4-5 predict partial and full ODEs, respectively, which are not seen in the observations. The
strength of the ODEs in February is underestimated by the model. A possible cause for this is an overestimation of halogen deposition over land, which can be seen in the comparison to satellite data and is discussed in section 3.3. Most of the model BrO capable of reaching Utqiaġvik can only be produced at Bering Sea during February due to a lack of sunlight in the northern regions. Since BrO over land is removed too quickly in the model, BrO can only be sustained through heterogeneous reactions while being transported from Bering Sea to Utqiaġvik by trajectories that go mostly over the sea ice.

In March, both simulations and observations agree in the occurrence of at least partial ODEs during most of the month whereas times without any ozone depletion at all are rare. Around March 4, the model predicts a partial ODE in simulations 2-4, whereas simulation 6 predicts a full ODE, neither of which is found in the observations. Four days later, all simulations

**Table 4.** Statistics at Utqiaġvik and Summit for the ozone mixing ratio for February 1, 2009 through May 1, 2009.

| Simulation | location | R [-] | average of the simulated variable [nmol/mol] | MB [nmol/mol] | RMSE [nmol/mol] |
|---|---|---|---|---|---|
| 1 | Utqiaġvik | -0.310 | 38.3 | 15.80 | 21.9 |
| 2 | Utqiaġvik | 0.617 | 27.6 | 5.09 | 12.1 |
| 3 | Utqiaġvik | 0.644 | 23.7 | 1.08 | 10.9 |
| 4 | Utqiaġvik | 0.454 | 29.5 | 6.97 | 14.3 |
| 6 | Utqiaġvik | 0.430 | 24.0 | 1.41 | 14.1 |
| 1 | Summit | 0.690 | 45.2 | -5.366 | 6.62 |
| 3 | Summit | 0.683 | 46.2 | -4.39 | 5.89 |

predict a partial ODE even though a full ODE is seen in the observations. The following ODE-free time period until March 13 is predicted in agreement with the observations, however, the full ODE on March 15 appears as partial ODE in all simulations, and
the simulations with enhanced emission find the partial ODE to continue for three more days. The ODE on March 19 is found in simulations 2-6. The simulations predict a near-full recovery of ozone levels over three days, which, however, is interrupted in the observations on March 21. The following ODE episodes are captured quite well by the simulations with an over-prediction of ozone levels on March 25 and March 28. ODEs around April 1, 14, and 18 are underestimated in the simulations, whereas all other ODEs and ozone regeneration episodes are predicted quite well. At the end of April, the observations find enhanced
ozone levels which are not captured by the model, not even by the simulations without the halogens. The enhanced ozone levels in the observations might be due to Arctic haze, i.e. enhanced photochemical ozone formation due to air pollution originating from lower latitudes. Walker et al. (2012) found that the decomposition of PAN, transported from lower latitudes or the upper troposphere to the arctic boundary layer, can account for up to 93% of the ozone production in the Arctic. The domain modeled in this work (see Fig. 1) does not consider the lower latitudes, so that the simulation itself cannot predict the production and
transport of Arctic haze. However, pollution from the lower latitudes might be correctly modeled by the MOZART-4 model and thus be present in the lateral boundary conditions. The model does not find these enhanced ozone levels, which suggests inaccuracies in the MOZART-4 boundary conditions. Simulation 3 finds a partial ODE on April 29, which is not present in the observations. The other simulations also find a slight decrease in the ozone mixing ratio, however, for these simulations, the BrO levels are not predicted to be large enough for an ODE to happen. Summarizing the entire period of three months,
simulation 1 shows two ODEs where none were observed. Twenty-two ODEs are identified in the observations, half of which are found by simulation 2. Simulation 3, however, identifies four additional ODEs compared to simulation 2 which were not found in the observations. Simulation 3 misses only six of the 22 observed ODEs.

     The results of simulation 6 differ strongly from the other simulation results starting mid March and the correlation coefficient R of 0.435 compared to simulation 2 with R = 0.62. The RMSE is 14.1 nmol/mol compared to 12.1 nmol/mol. The mean
bias is improved, but this is simply due to the enhanced emissions, resulting in more ODEs and not due to actually predicting the ODEs better. All statistics are worse compared to simulation 3. As discussed previously in this section, simulation 6 predicts meteorology much worse due to the lack of nudging, which also leads to wrong predictions in the ozone mixing ratio. As can

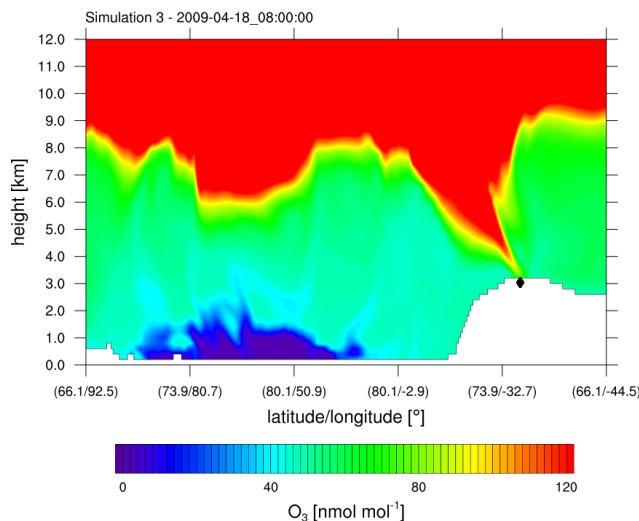

**Figure 6.** Ozone mixing ratio obtained from simulation 3 ($\beta = 1.5$) against height along a horizontal line through Summit, cf. Fig. 1. A tropopause fold reaches Summit (♦).

be seen in the correlation plots, simulations 2 and 4 rarely find ODEs were there are none in the observations. There is a notable accumulation of points in all 4 simulations at ozone mixing ratios of about 30–40 nmol mol$^{-1}$ for both the observations and the model. In this range of ozone mixing ratios, both the model and observations do not show any ODEs. Halogen chemistry, which has large uncertainties regarding the chemical reactions and the source of bromine, is less important in this case, which explains the high density of points in this regime. This accumulation is denser for simulations with weaker bromine emissions, since those simulations predict ODEs less often which do not exist in the observations. There is an additional accumulation of points around an ozone mixing ratio of zero in both model and observations for simulations 2-4, which are ODEs found by both model and simulation. This accumulation is less dense for simulation 4 compared to simulations 2 and 3. Simulation 4 performed worst regarding both mean bias and RMSE. In simulation 4, there is an accumulation of points at around modeled ozone values of 30 nmol mol$^{-1}$ and observed ozone values of zero, which are the missed ODEs by the simulation which suggests an underestimation of the occurrences of ODEs. Simulation 4 with a strongly enhanced $\beta = 2.0$ but a reduced bromine emission due to direct bromide oxidation by ozone during daytime ($\Phi = 0.1$) suggests that the bromine explosion mechanism alone is insufficient to properly predict the bromine production.

Simulations 2-4 and 6 reproduce ozone levels and ODEs much better than simulation 1, where the mean bias is smaller by at least 9 nmol/mol. For simulation 3, all statistics are improved compared to the base simulation 2, with both the correlation and RMSE being only slightly better and the mean bias being about 80% smaller (1.1 nmol mol$^{-1}$ vs. 5.1 nmol mol$^{-1}$) than in simulation 2. Figure 5 shows a strong increase in the number of ODEs that occur in the model but not in the observations, which explains the strongly improved mean bias while the other statistics only improved slightly.

At Summit, ODEs were found by none of the simulations and not in the observations which lack data for April 29 as can be seen in Fig. 4. The differences between a simulation without halogens and with halogens are negligible. Ozone mixing ratios

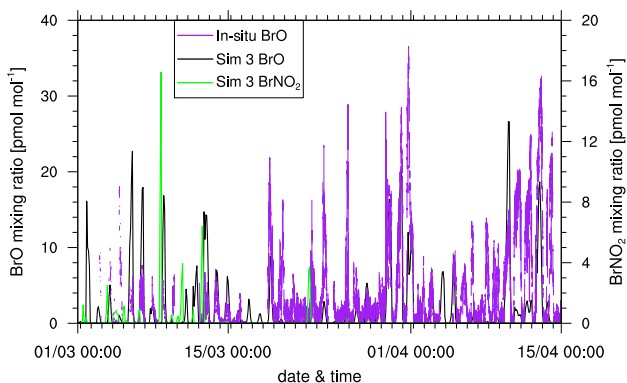

**Figure 7.** Comparison of modeled BrO and in-situ observations of BrO at Utqiaġvik (Liao et al., 2011) and modeled $BrNO_2$; the numerical results are for simulation 3.

are under-predicted with a mean bias of -4.3 nmol/mol for simulation 2. This is in contrast to Utqiaġvik, where ozone was generally over-predicted. In April, ozone levels at Summit are found to exceed 60 nmol $mol^{-1}$ for several time periods in the

395 observations. This is probably due to the high elevation of 3,200 m above sea level of Summit in contrast to Utqiaġvik. At Summit, the time with the highest ozone level which occurs on April 18 is found by the model. The high ozone mixing ratio in the model is due to stratospheric ozone, reaching the troposphere due to a tropopause fold event as shown in Fig. 6. The other time periods of enhanced ozone levels found by the observations may also be due a tropopause fold or possibly Arctic haze events.

Figure 7 shows modeled $BrNO_2$ and BrO of simulation 3 and in-situ observations of BrO (Liao et al., 2012) at Utqiaġvik. In order to improve the comparability of the observed data with a 10 min resolution and the model results which were saved every two hours, a seven-point moving average is applied on the observations, taking the average of the time point under consideration and three time points prior and after that time point. Modeled BrO is under-predicted with a mean bias of -1.65 pmol $mol^{-1}$ and a correlation of 0.472 is found. In early to mid March, BrO is less under-predicted with an over-prediction of BrO for some days. For most of these days, enhanced BrO levels are due to $NO_x$-catalyzed release of reactive bromine. $NO_x$ is emitted at Prudhoe Bay and can then produce $N_2O_5$, which further releases $BrNO_2$ on FY ice via the heterogeneous reaction

$$N_2O_5(g) + Br^-(aq) + H^+(aq) \longrightarrow BrNO_2(g) + HNO_3(g).$$

400 $BrNO_2$ then can photolyse to Br, which may further release bromine on FY ice through the bromine explosion mechanism. In the current model, the above heterogeneous reaction is the only source of $BrNO_2$, so that any enhanced mixing ratios of $BrNO_2$ at Utqiaġvik can be attributed to polluted air from Prudhoe Bay producing bromine on FY ice through the heterogeneous reaction with $N_2O_5$. As can be seen, for many of the days in early March, there are enhanced $BrNO_2$ mixing ratios preceding large BrO levels. Enhanced modeled BrO on February 14, 17, and 20, see Fig. 4, are coincident with large $BrNO_2$ mixing ratios

405 caused by polluted air from Prudhoe Bay which are transported over sea ice. A similar phenomenon was found by Simpson et al. (2018), who discovered large BrO concentrations in February 2017, which are attributed to nighttime photolabile bromine

production, possibly by $N_2O_5$, over sea ice. These photolabile species may be transported to lower latitudes where they might be photolyzed. A further discussion of modeled $N_2O_5$ can be found in section F of the supplement. From the end of March to April 15, however, the mixing ratio of modeled BrO is smaller, whereas the $BrNO_2$ mixing ratio drops to almost zero. Due to the higher temperature and stronger sunlight, $N_2O_5$ becomes less stable and its mixing ratio drops, suppressing bromine production due to $N_2O_5$. At the same time, observed BrO mixing ratios strongly increase. The under-prediction of modeled BrO for these later dates is likely due to a general under-prediction of bromine near coastal regions and on land, which will be further discussed in the following sections.

## 3.2   Vertical ozone and temperature profile at Utqiaġvik

Ozone sonde sounding data (Oltmans et al., 2012) produced near Utqiaġvik are used to validate vertical ozone profiles. Measured ozone and potential temperature for the upward flight of the sonde in the first 2 km are shown in Figs. 8 and 9 together with the simulation result of the column of the nearest grid cell. The simulation result is interpolated linearly in time to the starting time of the sonde flight.

Figures 8 and 9 show vertical profiles at Utqiaġvik for various dates. For March 14, the model fails to find the shallow surface inversion (boundary layer height smaller than 50 m) possibly due to a lack of vertical resolution. The boundary layer height of about 350 m in the observation is over-predicted by approximately 200 m by the model, which might also partially explain the finding of a partial ODE by the model instead of a full ODE as seen in the observations. For this day, simulation 3 performs slightly better than simulation 2. Two days later, both the observations and the simulations show partial ODEs. Simulation 2 predicts the ozone profile very well. The temperature profiles are quite different, however, both model and observations show an inversion at a similar, low height. For March 22, the enhanced emission case correctly predicts a full ODE, capturing both ozone and temperature profile quite well. The model is however unable to capture a surface inversion. On April 15, a surface inversion with a second inversion at approximately 500 m is found in the observations. The MYJ PBL scheme also predicts a surface inversion, however it fails to predict the second inversion properly, as can be seen by the lack of a second ozone plateau. While the model is unable to capture the complex boundary layers perfectly, the ozone profiles shows many similarities to the observed profile. For a better prediction, more grid levels closer to the surface and improvements to the PBL schemes might be needed. Even that, however, might not be sufficient, since PBLs in the Arctic can be influenced by very small-scale structures such as open leads, which were found to play an important role in the ozone recovery after an ODE due to down-mixing of ozone-rich air from the free troposphere and which would require high-resolution sea ice data. Additionally, an accurate modeling of surface inversions might require very high vertical resolutions which are difficult to obtain in a synoptic scale simulation.

Figure 10 shows modeled vertical BrO profiles convoluted with the MAX-DOAS averaging kernel from March 28, 2009 to April 16, 2009 at Utqiaġvik in comparison to BrO measured with a MAX-DOAS instrument (Frieß et al., 2011). The time range from February 26, 2009 to March 27 is illustrated in section E of the supplement, Fig. S4. BrO from the same observation dataset is shown in Figs. 8 and 9. On days with good visibility, the observed data is sensitive for the first 1-2 km. As can be seen, model and observations agree on most dates on the presence of BrO. However, modeled BrO tends to be elevated in

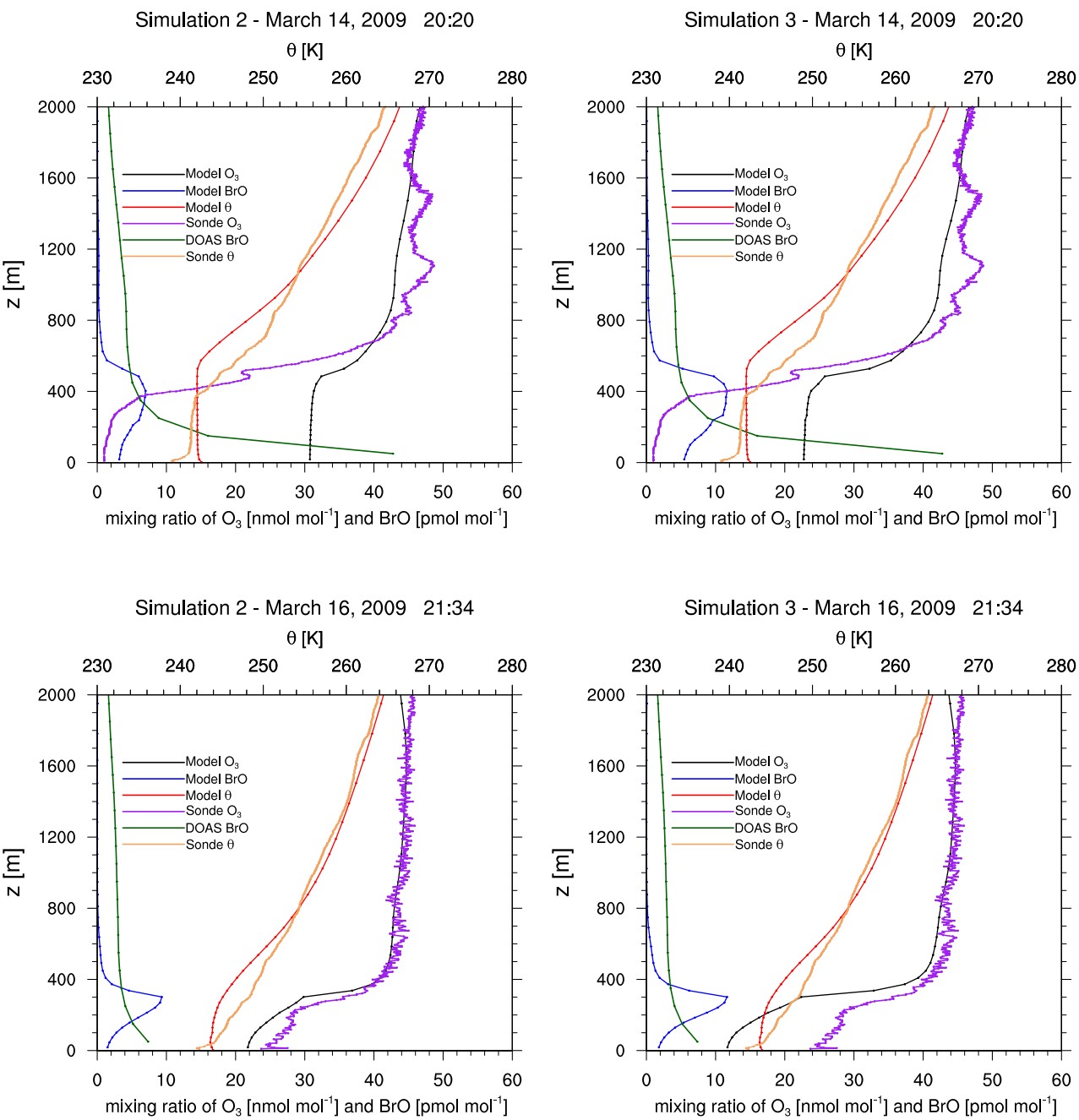

**Figure 8.** Vertical profiles of measured and modeled ozone, of potential temperature $\theta$, and of BrO at Utqiaġvik on March 14 (top) and on March 16 (bottom), 2009. Measurements are from upward flights using ozone sondes (Oltmans et al., 2012) and DOAS measurements (Frieß et al., 2011).

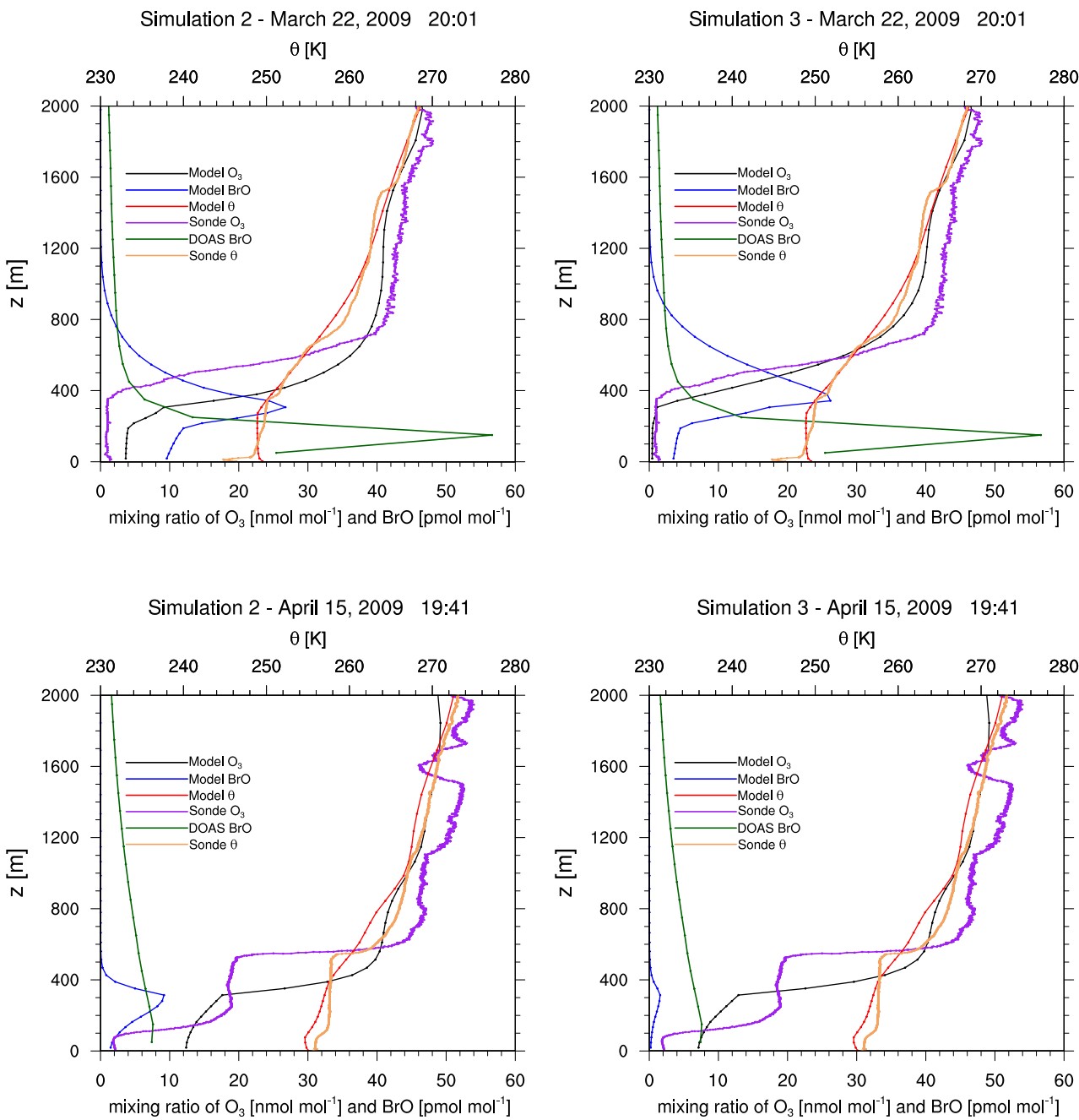

**Figure 9.** Vertical profiles of measured and modeled (simulation 2 (left) and simulation 3 (right)) ozone, of potential temperature $\theta$, and of BrO at Utqiaġvik on March 22 (top) and April 15 (bottom), 2009. Measurements are from upward flights using ozone sondes (Oltmans et al., 2012) and DOAS measurements (Frieß et al., 2011). On April 15, only the observed BrO mixing ratio in the lowest 100 m is accurate due to very poor visibility.

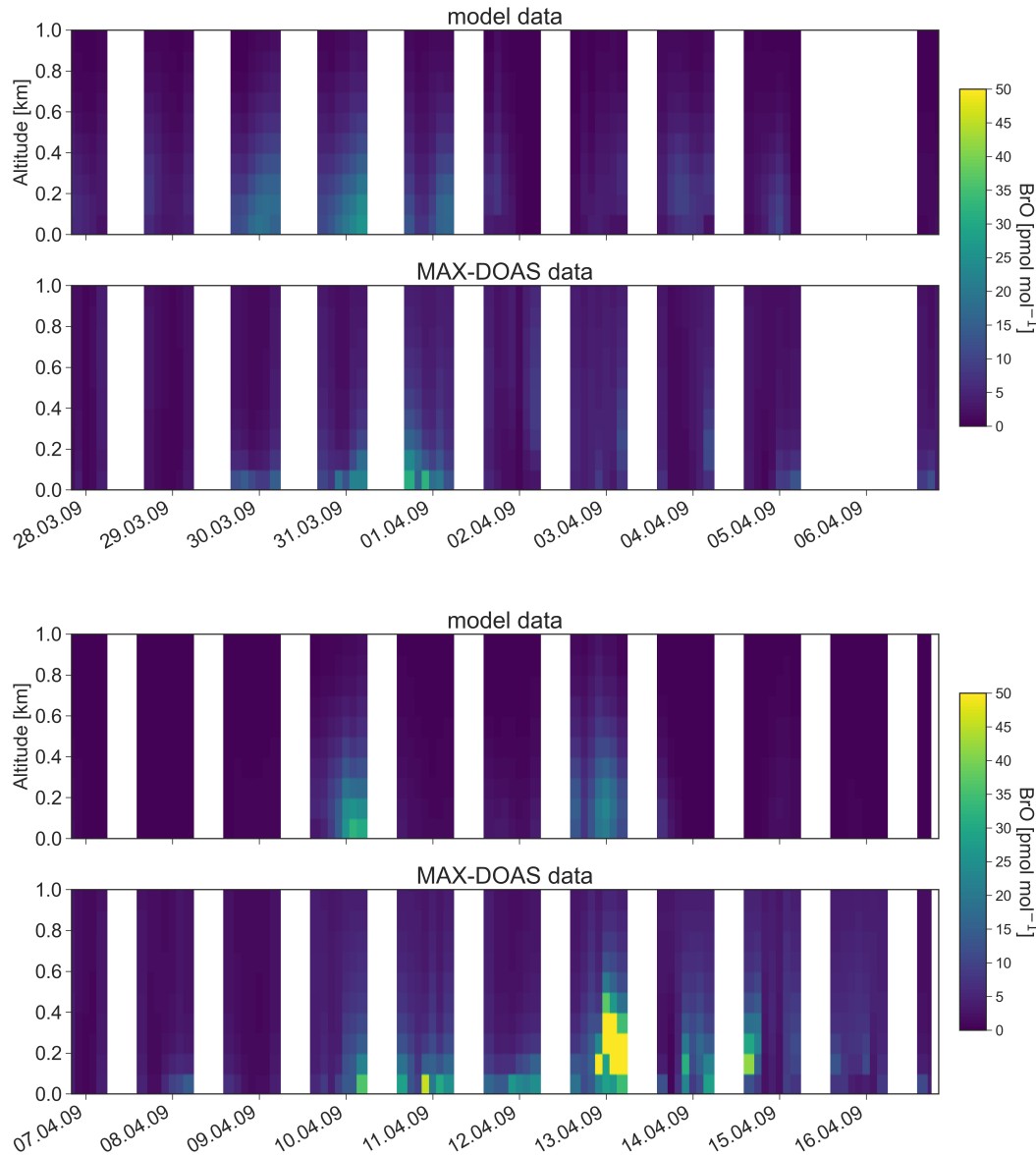

**Figure 10.** Vertical profiles of BrO from March 28, 2009 to April 16, 2009 at Utqiaġvik. Top: Modeled BrO convoluted with the MAX-DOAS averaging kernel, Bottom: BrO observed with MAX-DOAS

comparison to the observations, which can be seen for all days shown in Figs. 8 and 9 and on March 31 and on April 01 and 10 in Fig. 10. This is likely due to an underestimation of bromine emissions over snow-covered land, which is also discussed in the next section. Since the model assumptions only allow for partial recycling of bromine over land but not for new emissions, in the lowest grid cells, bromine is lost due to depositions, which results in the elevated modeled BrO profiles. On March 9 and

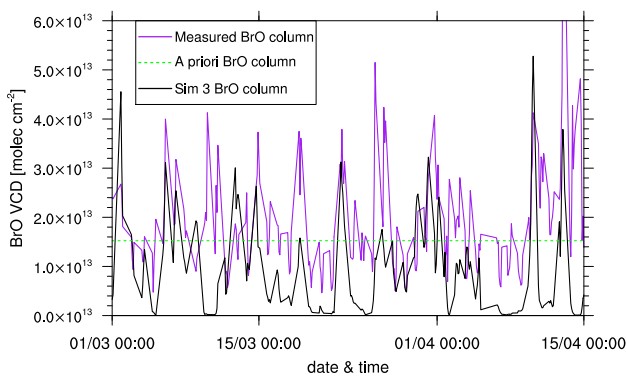

**Figure 11.** Comparison of modeled (simulation 3) BrO VCDs and measured BrO VCDs from MAX-DOAS at Utqiaġvik (Frieß et al., 2011). Also shown is an a-priori BrO column for days with low visibility.

13, the model over-predicts BrO. The high BrO mixing ratio on those two dates is due to a heterogeneous reaction involving $N_2O_5$, see Fig. 7. Frieß et al. (2011) found correlations of the aerosol extinction and BrO, which led to the hypothesis that BrO is released in-situ during snowstorms. Currently, there is no model for blowing snow included, which may explain the under-prediction of modeled BrO at some days.

Figure 11 shows vertically integrated modeled (simulation 3) and measured BrO (Frieß et al., 2011) over the first 2 km. As can be seen, the BrO column is generally under-predicted by the model with a mean bias of $-0.98 \times 10^{13}$ molec cm$^{-2}$. This may partly be attributable to the under-prediction of BrO over land in the model, however, there seems to be an offset of around $5.0 \times 10^{12}$ molec cm$^{-2}$ in the measurements. A correlation of 0.427 is found.

### 3.3 Tropospheric BrO VCDs

GOME-2 satellite tropospheric BrO VCDs (Sihler et al., 2012) described in section 2.6.1 are compared with BrO VCDs evaluated from the numerical simulations. All satellite BrO orbits of the same day are averaged and plotted into one figure, where missing satellite data are neglected. Since stratospheric BrO is not generated in the present model, all BrO predicted by the model is of tropospheric origin. Thus, model BrO VCDs are calculated by integrating BrO concentrations vertically from the bottom to the top of the calculation domain. Simulation results are stored every two hours starting at 00:00 UTC. Each output is assigned a 60° segment of a circle with its origin at the north pole. The segment is centered on a longitude, conforming to GOME-2 orbits for that time. The BrO VCDs are averaged with their neighboring segments with a weight of unity at the center of the segment, and decreasing linearly to zero at the edge of the segment. This procedure is a linear time interpolation and smoothes the resulting model BrO VCDs. Figure 12 displays the simulated instantaneous BrO VCDs on March 8, 2009 16:00 UTC and 18:00 UTCO. On the left there are two of the twelve full BrO VCDs saved for each day, on the middle the corresponding 60° segment multiplied by a weight of unity at the center, which linearly decreases to zero at the edges of the segment. On the right, the added segments are shown. This procedure is done each day for all twelve timepoints.

Thus twelve segments, not just the two segments shown in Fig. 12, are added for the average of one day, covering the whole domain.

Figure 13 shows daily averages for the satellite data and simulations 2 and 3 on selected days. On March 8, both the model simulation and the observations show a high BrO VCD in Nunavut, including King William Island. However, the models predict BrO VCDs to be strongly concentrated in a small area whereas the satellite BrO cloud is spread out more and reaches deeper into the Canadian mainland. On March 15, both model simulations and satellite observations find a bromine cloud over the Laptev Sea, reaching to the Siberian land mass. The modeled BrO VCDs are more pronounced, with simulation 3 having a different distribution of BrO being less consistent with the observations than simulation 2. The enhanced emissions in simulation 3 cause a stronger ODE in that region, which in turn depletes BrO in the ozone depleted area. Ozone mixes back into the ozone-depleted area from the edges of an ODE, which allows BrO to form there which is the reason for the elevated BrO-levels seen at the edges of the ODE. The bromine cloud is predicted by the model to extend to Chuckchi Sea in a thin stripe, which is barely seen in the observation. In both model results, a small BrO cloud in Hudson Bay is found, which is more pronounced and less consistent with the observations for simulation 3. On April 13, a ring-like BrO structure can be seen north or Kara Sea. The BrO-free center of the ring is due to an ozone depletion. Both simulations correctly find a BrO-free area near the north pole. An enlarged ODE is predicted, resulting in a thinner ring more consistent with the observations. The model, however, under-predicts BrO clouds near the Alaskan coast and finds enhanced BrO VCDs on Greenland in contrast to the observations.

In summary, both simulations 2 and 3 appear to be successful in capturing the general structures. Some of the differences might be explained by a higher model resolution, 20 km $\times$ 20 km, compared to the satellite data with a resolution of 40 km $\times$ 30 km, resulting in more detailed structures in the model. Other differences might be explained by the already discussed errors in the meteorology and under-predicted of BrO over land discussed below.

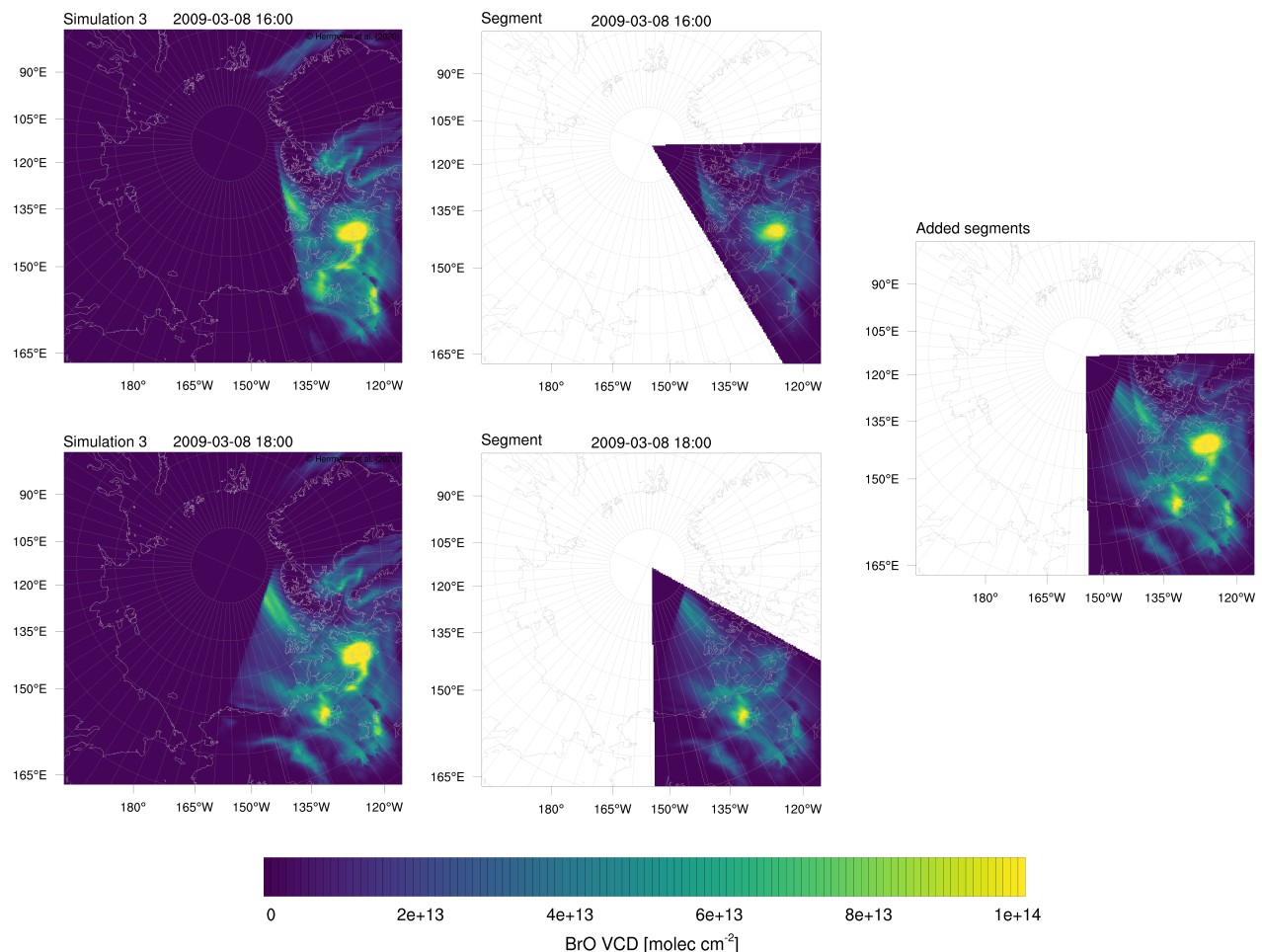

**Figure 12.** Illustration of the averaging of modeled BrO VCDs. Shown is March 8, 2009 16:00 and 18:00 UTC. Left: Full instantaneous BrO VCDs. Center: Corresponding 60° segment. Right: Added segments.

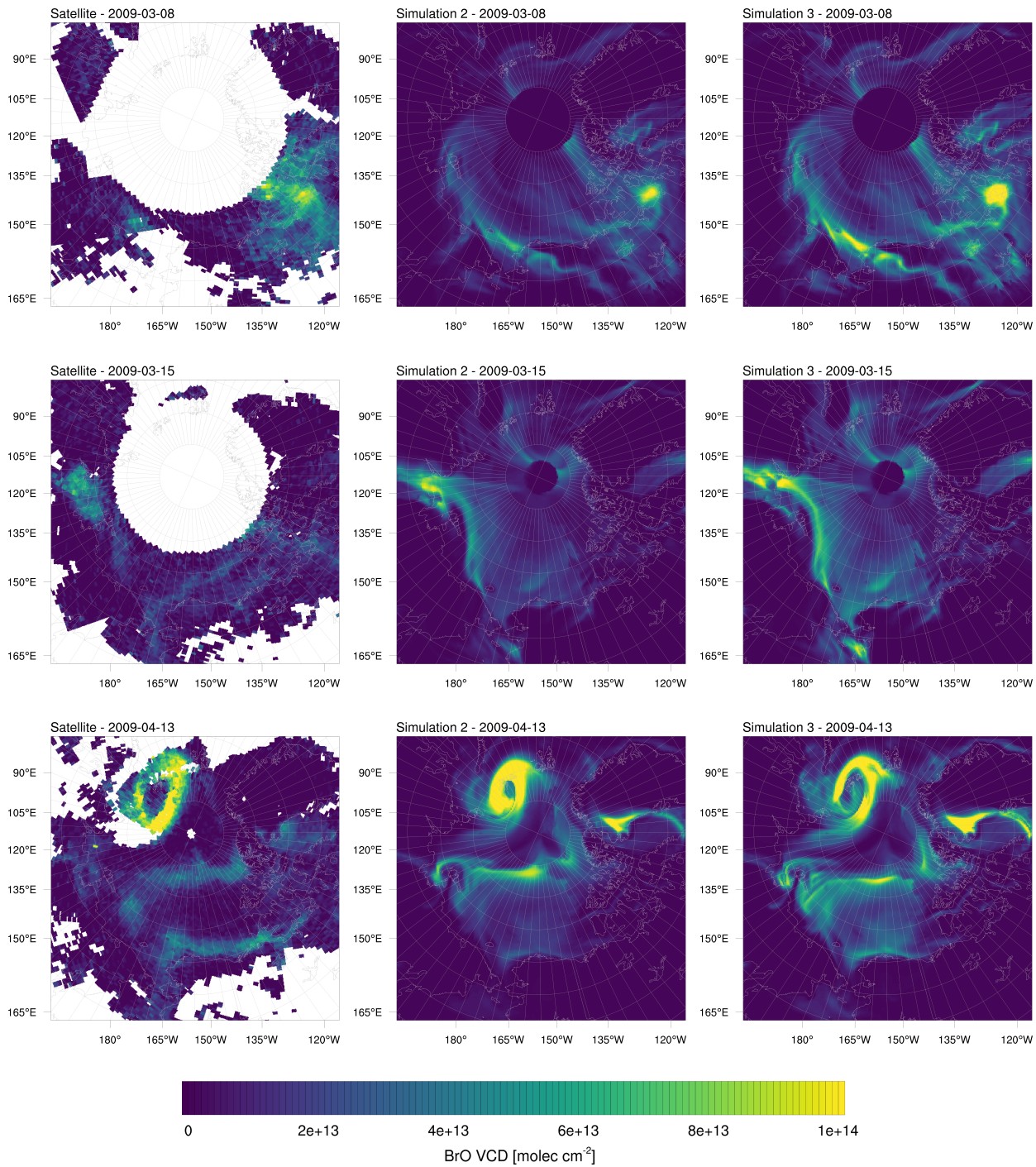

**Figure 13.** BrO VCDs on selected days in the year 2009. Left: Satellite measurements. Center: Simulation 2 ($\beta = 1.0$). Right: Simulation 3 ($\beta = 1.5$).

The uncertainties of the satellite data contribute to the differences between model and observations. According to Sihler et al. (2012), they are typically below 50%. Accordingly, differences in absolute values between model and satellite measurement might be to a substantial part be caused by measurement uncertainties. However, the spatial patterns found in the satellite data are hardly affected because measurements, which are strongly influenced by clouds (cloud shielding), are filtered out using the sensitivity filter of 0.5 for the air mass factor of the lowest 500 m (Sihler et al., 2012).

Figure 14 shows monthly averages for the satellite data and results of simulations 2 and 3. The simulations under-predict BrO over land and near coasts which is most likely due to the assumptions in the emission scheme. In the model, it is assumed that snow surfaces have no salt content, which makes depositions of bromine species (excess HOBr is lost) over land a sink, opposed to depositions over MY ice being neutral (excess HOBr is released as BrCl) and over FY (HOBr always releases $Br_2$) being a source for bromine in most cases. With a deposition velocity of 1 cm s$^{-1}$ and a boundary layer height of 200 m, bromine is removed at a timescale of approximately 5 hours over land by surface depositions and possibly even faster by depositions to aerosols. The assumption of zero bromide content of snow covering land or MY ice is of course an idealization and not always correct in reality (Simpson et al., 2005; Pratt et al., 2013; Peterson et al., 2018, 2019), contributing to the under-prediction of BrO over land mentioned in this paragraph. Future simulations should aim to find ways to incorporate the salinity, pH, and the Br$^-$/Cl$^-$ ratio of the snowpack, which where found to be important parameters for the production of $Br_2$ (Pratt et al., 2013; Wren et al., 2013; Peterson et al., 2019). BrO VCDs are also under-predicted near the boundaries, which is due to the value of zero of halogens at the boundary. The model over-predicts BrO VCDs at Baffin Bay and at most locations featuring FY sea ice with the exception of Bering Sea, probably due to its proximity to a domain boundary. The over-prediction over FY sea ice is not surprising with the assumption of unlimited BrO in FY sea ice. A relaxation of this assumption, e.g. by allowing finite salt content could solve the issues both, over snow covering FY ice by limiting the bromine emissions and over land, by allowing salt content of more than zero and storage instead of loss of deposited bromine. The model prediction for BrO in February is generally too small, which is probably due to a lack of sunlight at higher latitudes and the under-prediction of BrO over land. It should be noted that the satellite data is quite incomplete during February and biased towards the end of February, also due to a lack of sunlight necessary for satellite measurements in early February, whereas the model VCDs weights all of February equally.

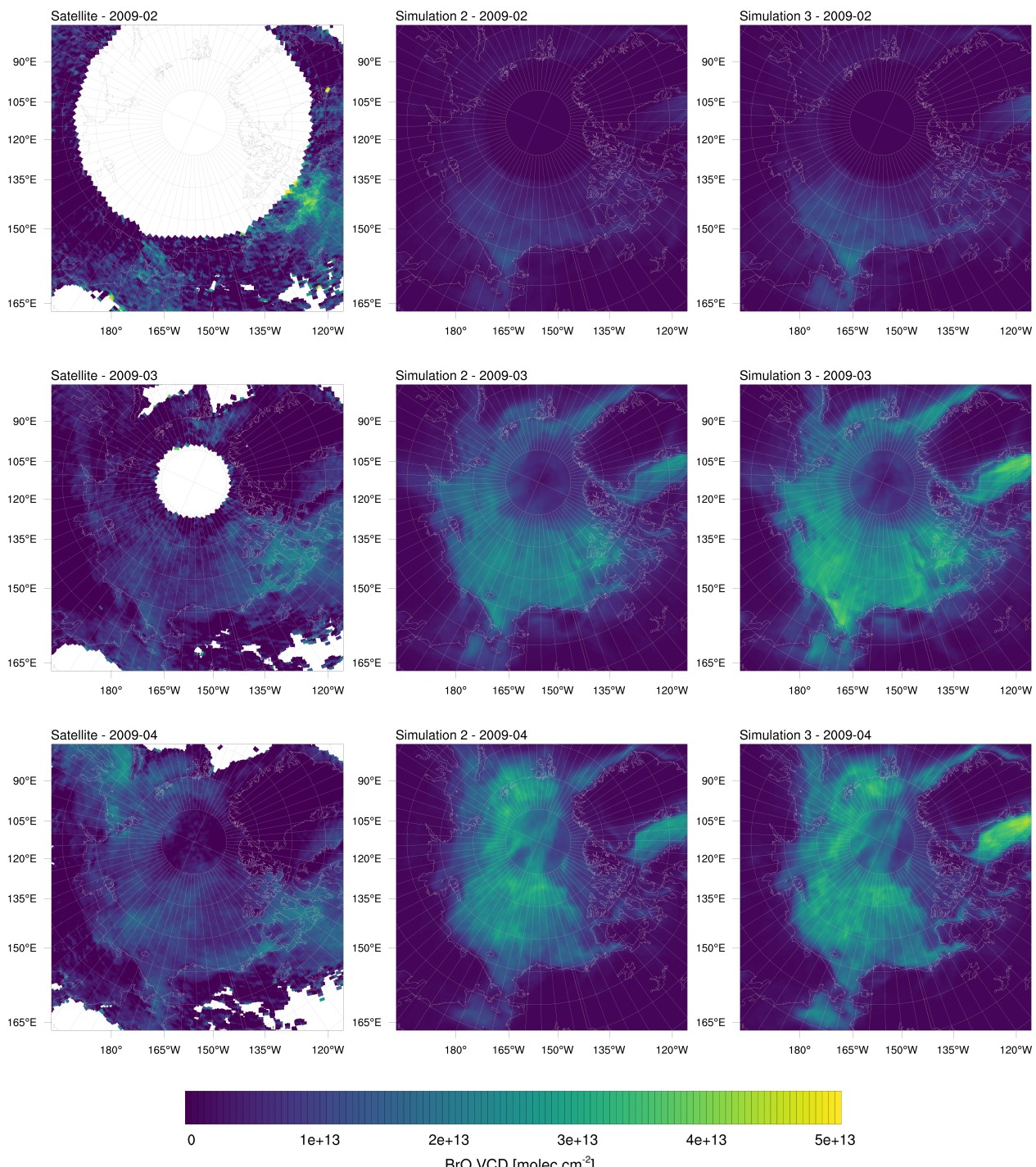

**Figure 14.** BrO VCDs in the year 2009 averaged over one month. Left: Satellite measurements. Center: Simulation 2 ($\beta = 1.0$). Right: Simulation 3 ($\beta = 1.5$).

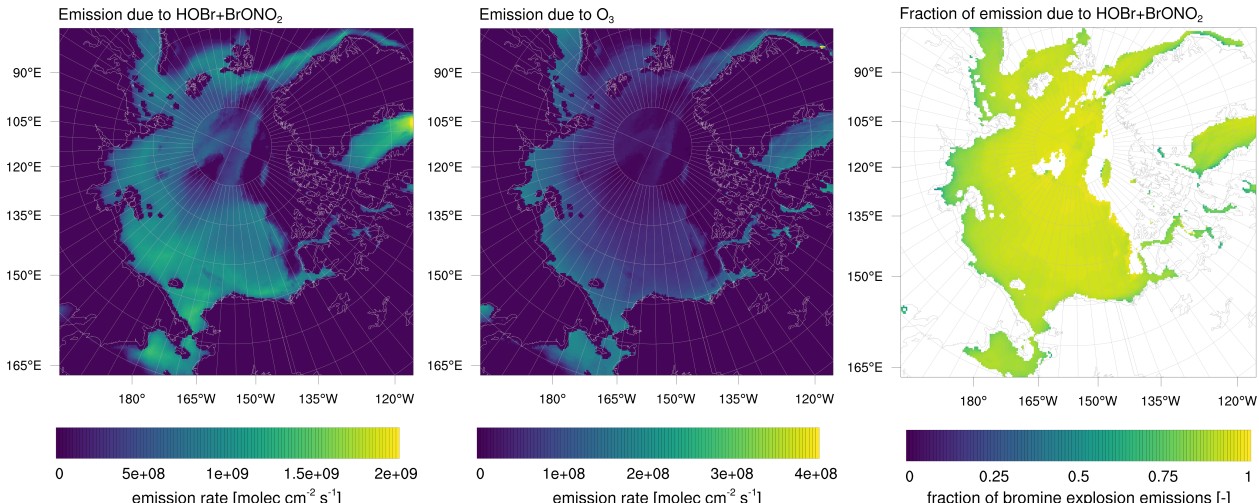

**Figure 15.** Emission rate of $Br_2$ due to $HOBr+BrONO_2$ (left) and due to bromide oxidation by ozone (center) from the snow surface for simulation 3, averaged over the complete simulation period. Ratio of $Br_2$ emissions due to HOBr and $BrONO_2$ to total $Br_2$ emissions on FY ice (right).

The emission rate of $Br_2$ due to $HOBr+BrONO_2$ and due to bromide oxidation by ozone from the snow surface averaged over the entire simulation period is shown in Fig. 15 for simulation 3. In Fig. 16, the production of $Br_2$ from the snow is shown at coordinates 178 W, 78 N plotted against time. The location has been chosen because it is over FY sea ice and is a strong

production site for the bromine that may affect ODEs at Utqiaġvik. As can be seen in these figures, most of the bromine is produced by HOBr, i.e. the bromine explosion mechanism, whereas the oxidation of bromide by ozone provides an initial seed of the bromine formation which then is enhanced by bromine explosion where $BrONO_2$ plays a smaller role than HOBr. Due to a lack of sunlight, bromine is produced only during the second half of February by the bromine explosion and after March 1, 2009, by the bromide oxidation due to ozone.

In the present parameterization, the latter strictly requires a SZA of less than $85°$ for a fast release, whereas the bromine explosion mechanism has a more continuous dependence on SZA. The $Br_2$ photolysis needed by both emission mechanisms requires relatively long-waved light and may thus occur even at SZAs slightly above $90°$. The bromine explosion additionally requires $HO_2$ in order to produce HOBr. $HO_2$ is mostly formed by a photolysis of various organic species with short-waved UV and thus occurs generally at smaller SZA, however, it can also be supplied by reactions involving organic compunds, $NO_x$

and/or OH or by their transportation from lower latitudes. Thus, in the present parameterization, the bromine explosion may occur locally at higher SZAs than the bromide oxidation due to ozone.

Emission rates of $Br_2$ from other studies are as follows. In February 2014, Custard et al. (2017) measured $Br_2$ fluxes of $0.07–1.2 \times 10^9$ molec $cm^{-2}$ $s^{-1}$ above the snow surface near Utqiaġvik with a maximum around noon. Wang and Pratt (2017) found snowpack $Br_2$ emissions of $2.1 \times 10^8$ molec $cm^{-2}$ $s^{-1}$ on March 15, 2012 and $3.5 \times 10^6$ molec $cm^{-2}$ $s^{-1}$ on

March 24, 2012 in a modeling study. Emission fluxes due to the bromine explosion (HOBr + $BrONO_2$) are typically between

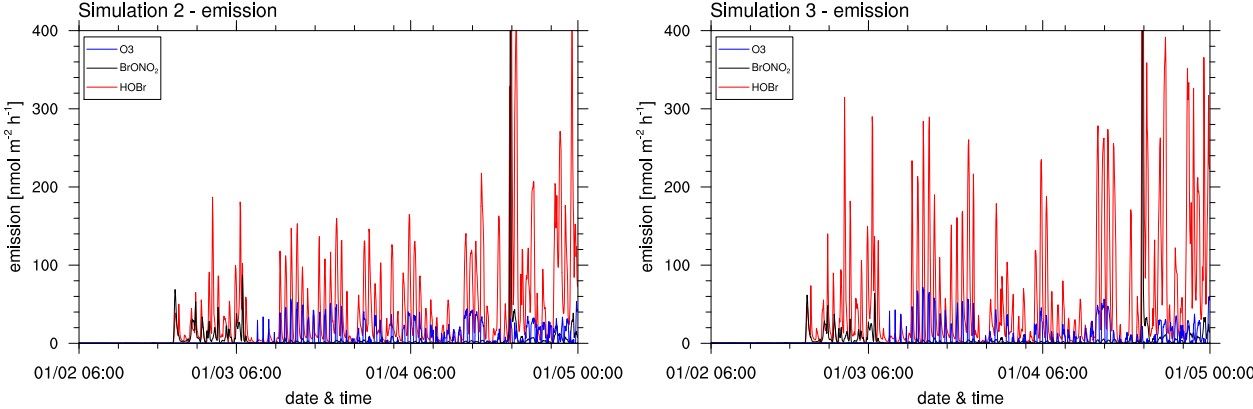

**Figure 16.** Emission rate of Br$_2$ due to HOBr and BrONO$_2$ and due to bromide oxidation by ozone from the snow surface at coordinates 178 W, 78 N for simulations 2 with $\beta = 1$ (left) and simulation 3 with $\beta = 1.5$ (right).

2-3$\times 10^9$ molec cm$^{-2}$ s$^{-1}$ (simulation 2) or 4-5$\times 10^9$ molec cm$^{-2}$ s$^{-1}$ (simulation 3) around noon and thus are on the higher end of the mentioned values. Bromide oxidation due to ozone, which plays the role of direct snowpack emissions in the present model, are rarely larger than $1\times 10^9$ molec cm$^{-2}$ s$^{-1}$ with an average of around $2\times 10^8$ molec cm$^{-2}$ s$^{-1}$ near Utqiaġvik, which compares quite well to the range found by Custard et al. (2017), while being larger than the values calculated by Wang and 
Pratt (2017).

For a simulation of three month, it should be expected that errors in the simulation pile up, especially considering the nonlinear stochastic nature of ODEs. The meteorological state should be consistent due to the data assimilation via nudging, however, the errors in the chemistry model could grow large over time. As an example, wrongly predicting an ODE probably causes a delay of an ODE at a later date due to the lack of O$_3$, reducing bromine emissions. A test for this is performing a new 
start of a simulation at a later date, where no ODEs occurred and in which the atmosphere is clean of bromine. For this purpose, simulation 5 was conducted, which is identical to simulation 3 except that the simulation starts on March 16 using ERA-Interim and MOZART-4 data as well as a near-zero bromine concentration, as described in section 2.5. These new simulation results are then compared to simulation 3 which started in February.

It is found that these two simulations become very similar after approximately 5 days, see Fig. 17 which shows the BrO 
VCDs. After approximately 8 days, the BrO VCDs become nearly indistinguishable. Average BrO concentrations in April are not shown here, but are also nearly identical for both simulations.

Reasons for the two simulations with different staring times to show so similar results after a few days is due to a combination of several factors. While there is no chemical nudging, the chemical boundary conditions strongly affect the simulation and act similar to a chemical nudging. Assuming a constant wind speed of 20 km/h (corresponding to approximately 5.5 m s$^{-1}$), 
a chemical species can be transported from a 2,000 km distant boundary to the center of the domain on a time scale of as low as four days. Due to the meteorological nudging, chemical boundary conditions are transported in the same way in both simulations.

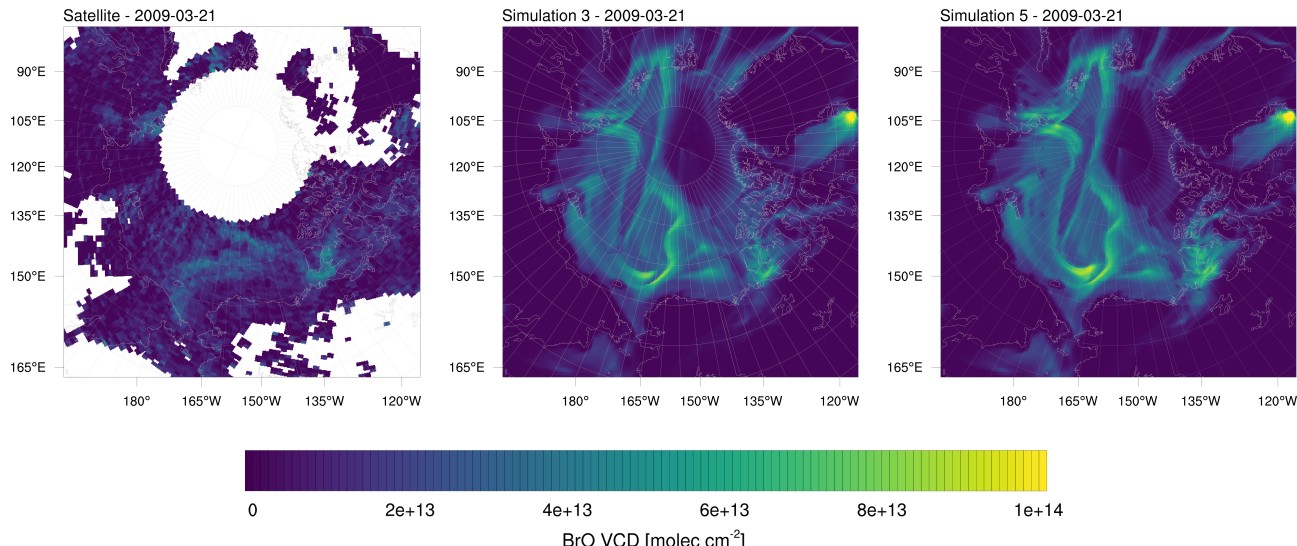

**Figure 17.** BrO VCDs on March 21, 2009 from observations (left) and simulations initiated on February 1 (simulation 3, center) and March 16 (simulation 5, right). The simulations differ only in the start time.

Chemistry boundary conditions transported over land or in the free troposphere behave similar in simulations 3 and 6, since several aspects of chemistry over land and in the free troposphere are nearly unaffected by the addition of halogen chemistry. Thus, chemical species coming from the lateral boundary condition will only be affected by the halogen chemistry once they reach the sea ice or are mixed into the boundary layer from aloft.

The emission of bromine due to bromide oxidation by $O_3$ is independent of reactive bromine mixing ratios and not of auto-catalytic nature as in the bromine explosion mechanism. While it is only responsible for small fraction of emitted bromine, it produces the initial bromine needed for a bromine explosion. The present emission scheme can be very fast, producing full ODEs in less than a day. All of these effects allow ozone coming from the lateral boundary condition to be depleted in a similar way in simulations 3 and 6 even with leftover bromine from a previous ODE.

## 4 Conclusions

Three-dimensional unsteady simulations of ozone depletion events in the Arctic from February 1, 2009 through May 1, 2009 have been performed using WRF-Chem. Simulations with different parameter settings are compared to observations from different sources at Utqiaġvik, Alaska and Summit, Greenland. A simulation using standard MOZART-MOSAIC chemistry without halogen chemistry resulted in an unrealistic ozone mixing ratio at Utqiaġvik, anti-correlating with observations and a strong bias for large ozone mixing ratios which demonstrates the impact of halogen chemistry on the prediction of ODEs.

Bromine may be emitted by the extended bromine explosion mechanism and/or oxidation of bromide by ozone directly from the sea ice. The reactive surface ratio $\beta$ accounts for non-flat surfaces such as snow/ice and controls the emission strength.

Both simulations with standard emission (simulation 2, $\beta = 1.0$) and a simulation with enhanced emission (simulation 3), $\beta = 1.5$ perform with correlations to observations of more than 0.6 at Utqiaġvik for both vertical ozone profiles and BrO VCDs. Enhancing the emission strongly improves the mean bias, whereas correlation and RMSE only improved slightly with enhanced emissions, which is likely due to an overestimation of BrO emissions which occur more frequently. Generally, ozone depletion at Utqiaġvik is somewhat under-predicted by both simulations. ODEs identified by the model that are not present in

the observations are rare: simulation 2 and 3 identify two and six ODEs, respectively. Simulation 2 finds half of the twenty-two observed ODEs whereas simulation 3 improves this prediction to more than two third of the observed ODEs. Iodine chemistry was neglected in this study, which may contribute to the under-prediction of ODEs at Utqiaġvik.

    At Summit, the observations and simulations agree in identifying no ODEs. A tropopause fold is found by the simulations at the end of April 2009 in agreement with the observations.

At Utqiaġvik, temperature is slightly over-predicted and wind speed slightly under-predicted both of which may contribute to an under-prediction of ODEs. BrO VCDs are found to be consistent with satellite observations. However, an under-prediction of BrO VCDs over land and an over-prediction of BrO VCDs over FY ice is apparent. A good qualitative agreement of modeled BrO with in-situ and MAX-DOAS measurements at Utqiaġvik was found, however the under-prediction of BrO over land was also apparent. This is probably due the assumptions of the emission scheme in the model: Snow covering FY ice is assumed

to have unlimited bromide content, resulting in an overestimation of BrO emissions, whereas snow over land has no halogen content, overestimating the removal of BrO. More realistic assumptions in a future study, such as an inclusion of snow pack emissions over land or a blowing snow parameterization, may improve the results.Emissions of bromine due to $N_2O_5$ were found to be important in February to mid March, but were of little relevance in the later months, since $N_2O_5$ becomes less stable with growing temperatures and sunlight intensity.

The direct emission of bromine due to bromide oxidation by ozone is found to be very important throughout the entire simulation, since it provides an initial seed of bromine which then triggers the bromine explosion. Simulation 4 with deactivated bromide oxidation by ozone under sunlight strongly reduces $Br_2$ emissions even though the value of $\beta$ has been set to 2.0. Therefore, simulation 4 is inferior to simulations 2 and 3 with a reduced overall prediction skill of ODEs. With even a larger emission rate, the bromine explosion mechanism alone does not produce enough BrO to explain the observations which is

likely due to a missing trigger of ODEs to provide the bromide oxidation by ozone. An alternative trigger of ODEs that may be worthwhile to study in future is the bromide oxidation by the hydroxyl radical.

    Meteorological nudging is found to be very important. A simulation with enhanced emissions by $50\%$ but disabled meteorological nudging (simulation 6) performs much worse compared to simulations 2 and 3. At Utqiaġvik, the prediction of meteorological variables such as temperature, for which the mean bias increased by a factor of three and the RMSE by a

factor of two, becomes worse during the simulation, in particular, the second half of the simulation has a strong bias to larger temperatures and a poorer skill for predicting ozone. Simulations 2 and 3 with $\beta$ equal to 1.0 and 1.5 respectively, are found to perform best where simulation 3 is somewhat superior to simulation 2 at the cost of an over-prediction of BrO at some times. It might be worthwhile to search for an optimal setting for $\beta$ in a future study.

In a follow-up study it is planned to simulate ODEs in the year 2019 for which the new TROPOMI BrO VCDs with a high
resolution of 5.5 km x 3.5 km are available. For this purpose, the grid resolution will be increased in order to allow for a
comparison of the more refined observation data.

*Code and data availability.*  The software code and data may be obtained from the corresponding author upon request.

*Author contributions.*  MH performed the simulations and wrote the paper draft. HS and UF contributed the observational data. TW provided
additional scientific support. UP and EG devised the methodology and supervised the project and EG revised the manuscript.

*Competing interests.*  The authors declare that they have no conflict of interest.

*Acknowledgements.*  The authors gratefully acknowledge funding by the Deutsche Forschungsgemeinschaft (DFG, German Research Foun-
dation) – Projektnummer 85276297 and through HGS Math-Comp. The authors acknowledge support by the state of Baden-Württemberg
through bwHPC and the German Research Foundation (DFG) through grant INST 35/1134-1 FUGG, allowing the authors to conduct sim-
ulations using the bwForCluster MLS&WISO Development. ERA-Interim data provided courtesy ECMWF. In-Situ and MAX-DOAS Data
were obtained from the OASIS (Ocean-Atmosphere-Sea Ice-Snowpack) 2009 campaign. GOME-2 level-1 data have been provided by
ESA/EUMETSAT. In-situ data for ozone at Utqiaġvik and Summit and meteorology at Utqiaġvik were obtained from the NOAA/ESRL
Global Monitoring Division (https://www.esrl.noaa.gov/gmd/). Ozone vertical profiles were obtained from the NOAA Earth System Re-
search Laboratory (ESRL,http://www.esrl.noaa.gov/gmd/obop/brw). OSI-403-c sea ice type data were obtained from EUMETSAT OSI SAF
(http://www.osi-saf.org/?q=content/global-sea-ice-type-c).

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
