# Peer review of "Time-dependent 3D simulations of tropospheric ozone depletion events in the Arctic spring using WRF-Chem"

_Atmospheric Chemistry and Physics, 2020_

## Referee Comment (RC1) · Anonymous Referee #1 · 7 Dec 2020

**1   general comments**

```
evaluating the overall quality of the discussion paper
```

- The manuscript is presenting an extension of halogen chemistry in the regional model WRF-chem and an implementation of a bromine release mechanism from ice and snow based on the work of ().

- The authors study bromine explosions and ozone depletion events (ODEs), respectively. They focus on one spring season (2009) at two sites, Utiqiaġvik, Alaska and Summit, Greenland. They discuss spatial extends (3 dimensions) as

well as temporal course (+1 dimension) of the events. The title of this manuscript is not reflecting this. A better title might be: *Spatio-temporal WRF-Chem simulations of ozone depletion events in the Arctic troposphere*.

- The authors use in situ measurements of $O_3$, temperature, wind speed, ozone sonde profiles, as well as GOME2 satellite retrievals of tropospheric $BrO$ to valid their model results.

- The manuscript is well structured.

- The scientific content of the manuscript is sound, there are only a few minor remarks.

- The language is overall concise but needs some refinement where the statements are not entirely clear.

**2   specific comments**

```
individual scientific questions/issues
```

Abstract

- L1–2: *"Tropospheric bromine release and ozone depletion events [...] are studied using the regional software WRF-Chem."* The term "regional software" is rather uncommon. It should either read as "regional model" or "mesoscale numerical weather prediction system with atmospheric chemistry module" based on the description given on the WRF web page
  (https://www.mmm.ucar.edu/weather-research-and-forecasting-model).

- L13: *"Meteorological nudging is found to be essential for a good prediction of ODEs [...]"* This finding is central to the manuscript. Given the close connection of the bromine release mechanism to weather conditions, this is not unexpected (). The verb "found" in this context, however, might have such a connotation. Maybe change "found" → "confirmed" to circumvent this interpretation.

Section 1

- L83–88: *"The salinity of the sea ice is also an important factor. [...]* $Br_2$ *emissions directly from the sea ice were not observed [...]."* There seems to be a contradiction in this paragraph. If no direct emission of bromine from sea ice is observed, how does its salinity affect emissions from a snow pack on top of it? The authors should elaborate on the logic of this paragraph as there seems to be a confusion between the roles of ice and snow in the release of bromine.

- L106–110: OASIS data is not mentioned any other place than here. If not used for the present model evaluation, what is the purpose of mentioning it? Are you intending to use them in follow-up studies as mentioned later on? You should make this clear, e.g. in Section 4.

Section 2

- L189: The authors state the use of ERA-interim. Why this choice? Other reanalysis data exists and in the meanwhile a global reanalysis of higher resolution ERA5 has been released, though there might be issues with these data especially in the Arctic. Taking into account the importance of nudging for predicting ODEs as one of the main points in this manuscript, could higher resolution ERA5 improve the model performance with respect to observations?

- L204–205: *"The initial mixing ratio of* HBr *and* Br$_2$ *are set to 0.3 ppt [...]. The mixing ratio of* CHBr$_3$ *is fixed to 3.5 ppt."* How sensitive is the model to the choice of these values? How well are they constrained by observations? Could actual heterogeneities in space and time explain the mismatch with observed ODEs? Have you considered oceanic emissions of very short-lived brominated species (CH$_2$Br$_2$, CH$_3$Br) as source terms in the model ()?

- *"Nudging is [...] inactive inside the boundary layer."* As nudging is one of the main points in this manuscript, this paragraph on nudging is a stub. How is the inactivity in the boundary layer realized (fixed height)? How strong is the nudging (nudging coefficients)? Is there a sharp transition between nudged and unnudged regimes of the atmosphere or is there a gradual relaxation of nudging towards the boundary layer? Are there systematic differences in boundary layer height between ERA-I and WRF which would affect the model results?

Section 3

- L221–232: The paragraph about the retrieval of BrO from GOME2 is too detailed for the purpose of this study – unless performed exclusively for this study. It should be shortened.

Section 4

- L247–251: *"The NOAA and ESRL Global Monitoring Division Surface Ozone measurements near UtqiaĄ̈avik and Summit [...]"* Are these observations independent or have they been assimilated into ERA-I?

- L267-278: *"[...] there is an overestimation of the temperature when it is cold, which is likely due to the lowest temperatures occurring during the spin-up time."* Unclear how these relate. May the authors elaborate on this?

- L277-278: *"Both wind speed and direction are predicted less accurately, which might result in wrong source locations or times of the occurrence of ODEs; this is likely to explain some of the differences between simulations and observations."* In other words, for a near real time prediction of ODEs one would need to assimilate observations. How large are the uncertainties on observed wind?

- L315–319: If lower latitude intrusion of polluted or ozone enriched air is a limiting factor, would nesting within a larger outer domain help improve the simulation result? Are there observations of "arctic haze" during these particular episodes available? Could these be used to improve the boundary conditions for the regional simulation?

- L353–355: *"[...] the time period with the highest ozone level is also found by the model which is due to stratospheric ozone, [...]"* The sentence is unclear and needs to be rewritten and probably split. Do the authors intend to say that the highest observed values of ozone at Summit are due to intrusion of stratospheric air masses in the course of tropopause folding events. This is correctly reproduced by the model?

- L374–375: *"[...] very small-scale structures such as open leads [...]"*Open leads could also lead to local emissions of brominated VSLS from the ocean.

- L375–376: *"[...]an accurate modeling of surface inversions might require very high vertical resolutions which are difficult to obtain in a synoptic scale simulation."* Could you achieve this by nesting?

- L406–409: *"Some of the differences might be explained by a higher model resolution [...] resulting in more detailed structures in the model. Other differences [...] errors in the meteorology [...]."* How does the uncertainty from satellite retrieval compare to the magnitude of divergence with modeling results?
- L503–505: *In a follow-up study it is planned to simulate ODEs in the year 2019 for which the new TROPOMI BrO VCDs with a high resolution of 5.5 km x 3.5 km are available. For this purpose, the grid resolution will be increased in order to allow for a comparison of the more refined observation data.* You should connect this statement with the earlier introduction of the OASIS dataset.

**3  technical corrections**

`purely technical corrections`

- L58: *"A further* $Br_2$ *release mechanism [...] was suggested, this was also found in a laboratory study [...]"* It would be better to split this sentence at the comma: "A further $Br_2$ release mechanism [...] was suggested. Evidence for this mechanism was found in a laboratory study [...]."

- L80: *"Temperatures [...] are likely to favour the occurrence of ODES [...]"* typo "ODES" → "ODEs"

- L104: *"[...] 3D air quality model GEM-AQ. [...] the EMAC model [...]"* These acronyms may need more explanation. For clarification: What are the major differences/improvments by using WRF-chem compared to the above mentioned models/simulations? GEM-AQ is much like WRF-Chem a weather prediction model with chemistry and aerosols able to run on different scales, while EMAC is a global chemistry climate model with a focus on middle atmosphere chemistry and dynamics.

- L232: *"[...] above a chosen sensitivity threshold of 0.5 are used.".* Units?

- L425: typo "weighs" → "weights"

- Fig. 6: Rainbow color color maps are generally depreciated for various reasons. First, it implies a distinct visual divergence of data at the edge between blue and green/yellow. This may lead to unintended misinterpretation() in some cases. Second, it is not colorblindness friendly. As the figure is purely used for illustration, chances of misinterpretations are low. Therefore, the authors may consider changing the color map, but it is not a must scientifically.

**References**

Toyota, K. and McConnell, J. C. and Lupu, A. and Neary, L. and McLinden, C. A. and Richter, A. and Kwok, R. and Semeniuk, K. and Kaminski, J. W. and Gong, S. -L. and Jarosz, J. and Chipperfield, M. P. and Sioris, C. E. *Analysis of reactive bromine production and ozone depletion in the Arctic boundary layer using 3-D simulations with GEM-AQ: inference from synoptic-scale patterns*, Atmos. Chem. Phys., 2011, vol. 11, no. 8, pp.3949–3979.

Lennartz, S. T. and Krysztofiak, G. and Marandino, C. A. and Sinnhuber, B.-M. and Tegtmeier, S. and Ziska, F. and Hossaini, R. and Krüger, K. and Montzka, S. A. and Atlas, E. and Oram, D. E. and Keber, T. and Bönisch, H. and Quack, B., *Modelling marine emissions and atmospheric distributions of halocarbons and dimethyl sulfide: the influence of prescribed water concentration vs. prescribed emissions*, Atmos. Chem. Phys., vol. 15, 2015, no. 20, pp. 11753–11772, 10.5194/acp-15-11753-2015,

D. Borland and R. M. Taylor Ii, *Rainbow Color Map (Still) Considered Harmful*, IEEE Computer Graphics and Applications, vol. 27, no. 2, pp. 14-17, March-April 2007.

---

## Referee Comment (RC2) · Anonymous Referee #2 · 30 Dec 2020

Herrmann et al present regional modeling of ozone and BrO in the Arctic spring troposphere.

The modeling focuses on Feb. – Apr. 2009 with two types of comparisons to observational data: 1) Arctic-wide maps of BrO vertical column densities, and 2) ground-level O3 at Utqiagvik, AK and Summit, Greenland. The time period was chosen to coincide with the OASIS field campaign at Utqiagvik, AK during which a large suite of ground-level measurements were made "for comparison with the numerical results" (as the authors state on Lines 107-109), but no comparisons are made, even to the authors' own near-surface BrO data at Utqiagvik, AK (Freiss et al. 2011, JGR). Comparisons to other available observational data, especially the available Br2, HOBr, and BrO mole ratios, from the campaign would yield improved understanding of the simulations. For

example, Figures 4 and 7 currently compare O3 between two simulations and observations, but only modeled BrO is shown. Since BrO was measured at the same location and time, this lack of comparison is a major weakness of the current paper that was pointed out in the quick review. The authors stated in their response: "It is planned to include the OASIS observations of Freiss et al. (2001) and Liao et al. (2012), but this will be done at the next review stage of the paper. . .We will also investigate further OASIS data, which may be compared to the modeling results." I consider this to be an essential and critical step for model evaluation. Not only will this make the manuscript much stronger, but it is a huge opportunity, especially given the suite of available data, including the vertically resolved BrO profiles (for the dates and location in Figure 7!) in Freiss et al. However, since this has not yet been completed, I cannot currently evaluate this.

The model is initiated with 0.3 ppt HBr and Br2 in the lowest 200 m, and CHBr3 is fixed to 3.5 ppt. Then Br2 emissions occur via BrONO2 and HOBr recycling on aerosols (for which the authors are encouraged to cite McConnell et al 1992, Nature (cited elsewhere in this manuscript) & Peterson et al 2017, ACP, "Observations of bromine monoxide transport in the Arctic sustained on aerosol particles" for observational support of this mechanism). The authors then include snow Br2 emission over sea ice via reaction of O3 with bromide and then claim "The bromine emission due to bromide oxidation by ozone is found to be important to provide an initial seed for the bromine explosion." (Abstract, Lines 10-12; also similarly stated on Lines 462-463 and 489-490) However, this is simply because of how the model is set up with this as the initial source, and it is not clear whether this setup and conclusion agrees with observations. Pratt et al. (2013, Nature Geoscience; which includes coauthor U. Platt) did not observe detectable Br2 production upon ozone reaction with snow bromide in the dark, whereas Br2 was produced when authentic snow was exposed to sunlight (without O3), showing efficient condensed phase formation of Br2. Addition of O3 initiates the traditional bromine explosion mechanism and further Br2 production (which is included in the model). The authors do not include condensed-phase Br2 production from the

snowpack and do state this in the manuscript as something for future work, but it is important when making conclusions statements that the model set-up and uncertainties be very clear, as this is otherwise misleading.

There is an opportunity presented by this work in Figures 12, 13, and associated discussion to compare the modeled snow Br2 emission rates to those previously measured by Custard et al (2017, ACS Earth & Space Chem.) during Feb in Utqiagvik and modeled by Wang & Pratt (2017, JGR).

In several locations in the Results & Discussion and Conclusions, the authors make statements that are not supported by the literature and modeling presented, especially when considering how the model is setup. Further the lack of comparison to BrO measurements at Utqiagvik weakens the results presented because the ground-level halogen chemistry at ground level cannot be properly evaluated. It is my hope that the authors will add OASIS measurements of Br2, BrO, and HOBr so that the model results can be better evaluated. Discussion that needs to be re-evaluated, in particular, includes:

Lines 287-289 and Lines 342-343: "A possible conclusion is that the bromine explosion mechanism is inefficient to explain ODEs in the Arctic, or the present bromine explosion scheme is incomplete for instance with respect to emissions of bromide containing aerosols due to blowing snow and/or regions of increased B such as frost flowers." "...the bromine explosion mechanism alone is insufficient to properly predict the bromine production." I'm guessing that the authors may be referring here to the Br2 production via O3 reaction with Br- (R10) as the "bromine explosion mechanism", but this would not be consistent with the literature, that refers to the bromine explosion as R1 + R5 + R6 + R8 +R9. Further, Wang et al. (2019, PNAS) showed, through simultaneous measurements of Br2, HOBr, BrO, Br atoms, and O3, that local ODEs could be quantitatively explained by the measured [Br], which would quantitatively show to be produced primarily by snow-phase reactions (condensed phase and heterogeneous). Further, the authors' statement is not in line with published literature and is further

weakened by the fact that the modeling in the current work is missing what has experimentally been shown as a major source of primary Br2. It needs to be acknowledged here that the model does not include sunlit condensed phase snowpack Br2 production (without O3), which Pratt et al (2013, Nat. Geosc.) found to be an efficient Br2 production mechanism, when Br2 was not detectable from dark reaction of O3 with snow, sea ice, or brine icicles (proxy for frost flowers). Sunlit condensed-phase Br2 production was also not observed for either sea ice or brine icicles, as predicted by Kalnajs and Avallone (2006, GRL), who stated "frost flowers are unlikely to be a direct source of atmospheric bromine."

Lines 300-302: The authors discuss here the transport of BrO from the Bering Sea to Utqiagvik, but it is important to remember that heterogeneous reactions are required in this scenario given the short lifetime of BrO (e.g., Platt and Honninger, 2003, Chemosphere; McConnell et al. 1992, Nature). Given their previous work, the authors are aware of this, but it needs to be clarified in the main text. Further, it is implied here that the model does not produce Br2 near Utqiagvik in February, but measurements in Utqiagvik in February 2017 showed local measurable Br2 snowpack fluxes (Custard et al 2017, ACS Earth & Space Chem). Further, U. Platt is a coauthor on the currently uncited, but highly relevant paper describing Jan.-Feb. BrO at Utqiagvik (Simpson et al. 2018, GRL).

Lines 315-316: The authors discuss here model inaccuracies associated with elevated O3, attributing this to haze or pollution. The prior modeling by Walker et al (2012, JGR, "Impacts of midlatitude precursor emissions and local photochemistry on ozone abundances in the Arctic") is relevant here and should be considered in interpreting the authors' model results (including the separate statement on lines 355-356). Also, the model here does not spatially resolve leads, the convection from which have been experimentally shown to control O3 recovery (end of the ODE) due to down-mixing from the free troposphere (Moore et al. 2014, Nature, "Convective forcing of mercury and ozone in the Arctic boundary layer induced by leads in sea ice"). The lack of spatial

resolution of leads and resulting impacts on meteorology (only briefly mentioned on line 375) and ozone should be considered in the discussion and interpretation of model results.

Lines 410-425 and Lines 483-488: This is good discussion, but it would further strengthen the discussion to incorporate observational literature to supports the interpretation here. For example, the model assumption that snow over land and near coasts have no salt content is inaccurate, as shown by Simpson et al (2005, GRL) and supported by BrO measurements by Peterson et al (2018, ACS Earth & Space Chem) and Pratt et al (2013, Nature Geosci.), which are cited elsewhere in the manuscript. Similarly, the discussion of FY sea ice vs MY sea ice as a source of bromine is presented, with measured snow [Br-] by Peterson et al (2019, Elementa) and includes a section of suggestions to modelers.

Additional Comments:

Lines 8-10: I suggest leaving the statements about the reactive surface ratios out of the abstract, as there isn't sufficient context here for the reader to understand what this physically means. It would be better to instead discuss other scientific results of the work, especially since this surface ratio seems to effectively be a model tuning parameter.

Lines 22-23: Note that Br atom reaction with ozone was directly observed by Wang et al. (2019, PNAS), so this can be stated more strongly here by inclusion of this reference.

Lines 32-33: Replace "ice" with "snow" to reflect current knowledge based on more recent direct measurements: Pratt et al. (2013, Nature Geoscience), Custard et al. (2017, ACS Earth & Space Chem.). Further, Pratt et al. (2013, Nature Geoscience) showed through direct measurements that sea ice did not produce detectable Br2. Further, while there is still much to be learned about Br2 production, as reflected in the sentence on lines 33-34, this paragraph primarily cites references from the 1990s and

does not discuss current knowledge gained from the last decade, thereby suggesting that less is known.

Line 40: The authors chose to designate heterogeneous reactions with "aq" above the arrow, but this is not a common notation. It would be clearer in the introduction to simply include the phases of each species as subscripts next to them in the equations so that it is clear to a novice reader which species are in the gas vs aqueous phase.

Line 45: The authors can strengthen this sentence by pointing to Pratt et al. (2013, Nature Geoscience), who directly showed the pH dependence of Br2 production through field-based experiments.

Line 57: Please clarify what is meant by "the surface layer" here. I believe the authors are discusses the snow grain surface, but this phrasing is also used for the atmosphere. Also, the authors should note, for clarity here, that sunlight is required for condensed-phase OH production.

Lines 61-67: References are needed for these sentences. The work of Thompson et al (2015, ACP, "Interactions of bromine, chlorine, and iodine photochemistry during ozone depletions in Barrow, Alaska") is highly relevant here, especially since that work focused on the OASIS field campaign.

Lines 75-76: The authors cite a 2007 review here for BrCl production via HOBr + Cl-. It is important to note that McNamara et al (2020, ACS Earth & Space Chem), who measured BrCl, showed that the Cl2 + Br- reaction is also a significant source of BrCl in the Arctic spring. However, despite mentioning these sources of Br atoms, Tables S3 and 4 do not include these reactions, suggesting that they are not in the model. Please clarify this in the text, as it is an additional uncertainty associated with the model setup.

Lines 80-81: Note that more recent work by Halfacre et al (2014, ACP) showed ozone measurements from five buoys across the Arctic and "no apparent dependence [of ODEs] on local temperature".

Line 89: Nasse et al 2019 (Atmos Meas Tech) appears to be the incorrect reference here, as it does not include Arctic ozone data.

Lines 93-94: Please connect this sentence about Thomas et al. (2011) to the halogen chemistry being discussed in this paragraph.

Figure 1: Provide the source of the sea ice data. Also label the latitude rings and especially where 88N is, and state in the caption that FY ice is assumed above this latitude, as stated in the text.

Lines 141-142: Note that iodine atom reaction with ozone has been shown to significantly increase the ozone depletion rate, even at low mole ratios (e.g. Raso et al. 2017, PNAS, showed 0.3 ppt of I2 to increase the initial rate of ozone depletion by 31% in a case study). Adding iodine chemistry to the model would be a significant task, so I think it is ok currently to not include. However, a greater acknowledgement of uncertainty is required here and should be mentioned again in the discussion and conclusions when comparing ozone levels between measurements and modeling.

Line 170 and Table S3 caption: Change "ice/snow" to "snow" here, since Line 191 states that all ice is assumed to be snow-covered.

Line 193: This is the only place that I saw N2O5 deposition mentioned as a source of Br2. This is intriguing and should be discussed in the results. What level of N2O5 is predicted by the model, and what fraction of the Br2 is predicted to be produced? I'm also confused because I don't see this heterogeneous reaction in Table S3. McNamara et al. (2019, Environ. Sci. Technol.) measured N2O5 in Utqiagvik during spring, which would provide a comparison point.

Lines 234-245: It is suggested to move this text and Table 2 to the Methods section, as results are not presented here.

Line 281 and Figure 4 caption: It is stated that "Figure 4 shows modeled and observed surface ozone and BrO at Utqiagvik" when measured BrO is not shown, and this is

misleading. It is my hope that the authors will rectify this by adding measured BrO (from the OASIS study) to Figure 4!

Lines 286, 341-343, and Lines 472-473: "This suggests a strong underestimation of BrO emission without a direct emission of BrO due to ozone." "...a reduced BrO emission due to direct bromide oxidation by ozone..." "BrO may be emitted by the extended bromine explosion mechanism and/or oxidation of bromide by ozone directly from the sea ice." These statements need to be rephrased as BrO is not directly emitted from snow or sea ice.

Table 4: Why is simulation 2 shown here for Summit, instead of simulation 3, which was deemed the best setup for Utqiagvik (since simulation 2 missed half of the ODEs at Utqiagvik)? This should be discussed.

Line 350: Fig. 5 doesn't show Summit results as stated here.

Figures 9-12, 14: Please make the latitude and land lines more visible to aid interpretation of the plots. Also, clarify in the Figure 9 caption what is meant by "cone".

Figure 12: Does this include only snow, or snow + aerosols? Please clarify.

---

## Author Comment (AC1) · 20 Feb 2021

**Authors' response to the reviewers**

The authors thank the reviewers for their valuable comments which lead to a great improvement of the present submission. We revised the manuscript where modifications of the text are marked in blue color. In particular, we address the comments in detail as follows.

**Referee #1**

**general comments**

**Reviewer:**
The authors study bromine explosions and ozone depletion events (ODEs), respectively. They focus on one spring season (2009) at two sites, Utqiaġvik, Alaska and Summit, Greenland. They discuss spatial extends (3 dimensions) as well as temporal course (+1 dimension) of the events. The title of this manuscript is not reflecting this. A better title might be: Spatio-temporal WRF-Chem simulations of ozone depletion events in the Arctic troposphere.

**Authors' Response:**

The current title is '3D simulations of tropospheric ozone depletion events using WRF-Chem'. We agree that a change of title is useful in order to emphasize the time dependence of the simulations and the location under consideration. However, the suggested title does not stress the three-dimensional nature of our model. We therefore prefer the new title 'Time-dependent 3D simulations of tropospheric ozone depletion events in the Arctic spring using WRF-Chem'.

**specific comments**

**Reviewer:**
L1–2:"Tropospheric bromine release and ozone depletion events [...] are studied using the regional software WRF-Chem. The term 'regional software' is rather uncommon. It should either read as 'regional model' or 'mesoscale numerical weather prediction system with atmospheric chemistry module' based on the description given on the WRF web page
(https://www.mmm.ucar.edu/weather-research-and-forecasting-model).

**Authors' Response:**
This is a valid point, we removed the word 'regional' and put instead '[...] are studied using the open-source software package WRF-Chem' in line 2 of the paper.

**Reviewer:**
L13: 'Meteorological nudging is found to be essential for a good prediction of ODEs [...]' This finding is central to the manuscript. Given the close connection of the bromine release mechanism to weather conditions, this is not unexpected (). The verb "found" in this context, however, might have such a connotation. Maybe change "found" → "confirmed" to circumvent this interpretation.

**Authors' Response:**
We agree with the reviewer that nudging is important, we therefore changed the sentence to read 'Meteorological nudging is essential for a good prediction of ODEs over the three-month period.' The idea of the present work is not to try to make meteorological predictions (which would not be meaningful anyway on the timescale of a few months) but rather to model chemistry under

meteorological conditions prevailing over a particular period of time. We now specify this in section 2.1 of the revised paper.

**Reviewer:**
L83–88: "The salinity of the sea ice is also an important factor. [...] $Br_2$ emissions directly from the sea ice were not observed [...]." There seems to be a contradiction in this paragraph. If no direct emission of bromine from sea ice is observed, how does its salinity affect emissions from a snow pack on top of it? The authors should elaborate on the logic of this paragraph as there seems to be a confusion between the roles of ice and snow in the release of bromine.

**Authors' Response:**
We like to thank the reviewer for this suggestion to clarify this paragraph. Abbatt et al. (2012) provide explanations for how the salinity of FY ice affects the snow pack. Essentially, brine is much more likely to form on FY ice than on MY ice. It then migrates upwards to the snowpack via capillary forces. The text on line 90 is modified to read 'The age of the sea ice is also an important factor. Snow covering FY ice, which has more accessible salt than MY ice, is expected to be the main source of bromine.' On line 94, we now clarify that the observation of no direct emission of bromine from sea ice was found by Pratt et al. (2013): 'Pratt et al. (2013) did not directly observe $Br_2$ emissions from the sea ice, which is likely due to a higher pH of the sea ice due to buffering (Wren et al., 2012).'

**Reviewer:**
L106–110: OASIS data is not mentioned any other place than here. If not used for the present model evaluation, what is the purpose of mentioning it? Are you intending to use them in follow-up studies as mentioned later on? You should make this clear, e.g. in Section 4.

**Authors' Response:**
We like to thank the reviewer for this suggestion. We decided to add in-situ data although they only provide information for one column of grid cells. Therefore, in the revised version, in-situ BrO observations at Utqiaġvik (Liao et al., 2012) are incorporated into section 3.1 and DOAS observations of vertical BrO (Frieß et al., 2011) are added to section 3.2 of the revised manuscript, which were not part of the submitted discussion paper. Both sets of observations were made as part of the OASIS campaign.
The following discussion was added to the end of section 3.1 of the revised paper, regarding the in-situ BrO observations and $NO_x$-catalyzed release of reactive bromine:
'Figure R1 (Fig 7 in the revised paper) shows modeled $BrNO_2$ and BrO of simulation 3 and in-situ observations of BrO (Liao et al., 2012) at Utqiaġvik. In order to improve the comparability of the observed data with a 10 min resolution and the model results which were saved every two hours, a seven-point moving average is applied on the observations, taking the average of the time point under consideration and three time points prior and after that time point. Modeled BrO is under-predicted with a mean bias of -1.65 pmol $mol^{-1}$ and a correlation of 0.472 is found. In early to mid March, BrO is less under-predicted with an over-prediction of BrO for some days. For most of these days, enhanced BrO levels are due to $NO_x$-catalyzed release of reactive bromine. $NO_x$ is emitted at Prudhoe Bay and can then produce $N_2O_5$, which further releases $BrNO_2$ on FY ice via the heterogeneous reaction

$$N_2O_5(g) + Br^-(aq) + H^+(aq) \longrightarrow BrNO_2(g) + HNO_3(g).$$

[revised manuscript text omitted]

**Reviewer:**

L189: The authors state the use of ERA-interim. Why this choice? Other reanalysis data exists and in the meanwhile a global reanalysis of higher resolution ERA5 has been released, though there might be issues with these data especially in the Arctic. Taking into account the importance of nudging for predicting ODEs as one of the main points in this manuscript, could higher resolution ERA5 improve the model performance with respect to observations?

**Authors' Response:**

The ERA-Interim Reanalysis was found to perform well in polar regions in various studies (e.g. Bracegirdle et al., 2012, Bromwich et al., 2016) and was successfully used in various modeling studies in polar regions (e.g. Hines et al., 2015, Cai et al., 2018), which is why it was chosen in the present study. When the work presented in this manuscript began, ERA-5 was not yet readily available for the year 2009, and we decided against changing to ERA-5. Wang et al. (2019) found a smaller warm-temperature bias (+3.4°C versus +5.4°C) for ERA-Interim in comparison to ERA-5, when the observed surface temperature was below -25°C. Graham et al. (2019) report a better agreement of observed and modeled temperature inversion over sea ice for ERA-Interim in several cases. Graham et al. (2019), however, found an improvement in ERA-5 for the wind fields. Overall, it appears that there is no compelling reason to prefer ERA-5 over ERA-Interim data. The initial part of this discussion has also been added to section 2.1 of the revised paper.

**Reviewer:**

L204–205: "The initial mixing ratio of HBr and $Br_2$ are set to 0.3 ppt [...]. The mixing ratio of $CHBr_3$ is fixed to 3.5 ppt." How sensitive is the model to the choice of these values? How well are they constrained by observations? Could actual heterogeneity in space and time explain the mismatch with observed ODEs? Have you considered oceanic emissions of very short-lived brominated species (CH2Br2,CH3Br) as source terms in the model ()?

**Authors' Response:**

The bromide oxidation of ozone in the dark for an ozone deposition velocity of 0.01 cm $s^{-1}$, a boundary layer height of 200 m, an emission probability of $\Phi = 0.001$, and 40 nmol $mol^{-1}$ ozone will release approximately 2 pmol $mol^{-1}$ $Br_2$ on FY ice per day. This emission rate is assumed to prevail for all simulations with active halogen chemistry. The chosen initial halogen concentrations and the fixed mixing ratio of $CHBr_3$ thus are irrelevant. We did not yet consider oceanic emissions of very short-lived brominated species. This discussion has been added to

section 2.5 (description of the initial conditions) of the revised manuscript.

**Reviewer:**
"Nudging is [...] inactive inside the boundary layer." As nudging is one of the main points in this manuscript, this paragraph on nudging is a stub. How is the inactivity in the boundary layer realized (fixed height)? How strong is the nudging (nudging coefficients)? Is there a sharp transition between nudged and unnudged regimes of the atmosphere or is there a gradual relaxation of nudging towards the boundary layer? Are there systematic differences in boundary layer height between ERA-I and WRF which would affect the model results?

**Authors' Response:**
Nudging is turned off for all grid cells at altitudes that are lower than the PBL height, determined by the boundary-layer model (the Mellor-Yamada-Janjic scheme in this work). The nudging coefficient is $0.0003 \, \text{s}^{-1}$, which corresponds to a nudging timescale of one hour. Nudging in the boundary layer is not necessary because of the short time scale governing boundary layer meteorology, which is much better resolved by the time step of one minute in our WRF-Chem simulation in comparison to the six-hour time resolution of the ERA-Interim data. The boundary layer height estimated by the model is generally smaller than the boundary layer height estimated by ERA-Interim. WRF-Chem predicts boundary layer heights of less than 100 m over most of the sea ice in March, whereas ERA-Interim generally predicts boundary layer heights of more than 300 m above sea ice. In comparison to ozone sonde data at Utqiaġvik, the boundary layer height predicted by the model matches the measurements quite well with a slight tendency for an over-prediction of the modeled boundary layer height. We see no necessity for a change of the text in the revised paper.

**Reviewer:**
L221–232: The paragraph about the retrieval of BrO from GOME2 is too detailed for the purpose of this study – unless performed exclusively for this study. It should be shortened.

**Authors' Response:**
The reviewer made a good point. The paragraph about the retrieval, section 3, is shortened in the revised paper.

**Reviewer:**
L247–251:" The NOAA and ESRL Global Monitoring Division Surface Ozone measurements near Utqiaġvik and Summit [...]" Are these observations independent or have they been assimilated into ERA-I?

**Authors' Response:**
The ERA-Interim list of observations (`https://www.ecmwf.int/sites/default/files/elibrary/2010/11692-list-observations-assimilated-era-40-and-era-interim-v10.pdf`) does not mention the observatories at Utqiaġvik/Barrow and Summit, therefore we believe that the observations are independent.

**Reviewer:**
"[...] there is an overestimation of the temperature when it is cold, which is likely due to the lowest temperatures occurring during the spin-up time." Unclear how these relate. May the authors elaborate on this?

**Authors' Response:**

During the spin-up time of the model, we expect the modeling errors to be larger. Since coldest temperatures occurred in the first two weeks of the simulation, during the spin-up time, the errors for cold temperatures are larger. The overestimation of temperatures, however, may also be explained by the warm bias of the ERA-Interim reanalysis when it is colder than -25°C (+3.4°C), see above. We reworded lines 268-269 of the original manuscript to read: '[...] there is an overestimation of the temperature when it is cold, which is likely due to the lowest temperatures occurring during the spin-up time during which the modeling errors are larger compared to other times. The ERA-Interim Reanalysis is known to have a warm bias for temperatures below -25°C (Wang et al., 2019a), which may also explain the deviations.'

**Reviewer:**

L277-278: "Both wind speed and direction are predicted less accurately, which might result in wrong source locations or times of the occurrence of ODEs; this is likely to explain some of the differences between simulations and observations. "In other words, for a near real time prediction of ODEs one would need to assimilate observations. How large are the uncertainties on observed wind?

**Authors' Response:**

In the Barrow Meteorological Station (BMET) Handbook (`https://www.arm.gov/publications/tech_reports/handbooks/bmet_handbook.pdf`), it is mentioned that an instrument accuracy of 0.17 m s$^{-1}$ for wind speeds between 0.4 and 75 m s$^{-1}$, a 5.6° wind direction resolution, and 0.25°C instrument accuracy for temperatures between -65 and -20°C. This is mentioned and discussed in the revised paper, starting on line 93. The RMSE of modeled and observed fields is at least one order of magnitude larger than the mentioned accuracies and resolutions, so that an assimilation of observations is likely to improve the real-time prediction of ODEs.

**Reviewer:**

L315–319: If lower latitude intrusion of polluted or ozone enriched air is a limiting factor, would nesting within a larger outer domain help improve the simulation result? Are there observations of "arctic haze" during these particular episodes available? Could these be used to improve the boundary conditions for the regional simulation?

**Authors' Response:**

An outer nest, which would be a 15,000 km×15,000 km domain with the current setup should, together with a proper emission dataset, allow for a modeling of arctic haze. It might be necessary to start the outer nest simulation at an earlier date to allow for the transport from the mid latitudes to the arctic region. Using observations of 'Arctic haze' is a good idea, however, we have no indications of any observations of arctic haze in the time interval of our model calculations.

**Reviewer:**

L353–355: "[...] the time period with the highest ozone level is also found by the model which is due to stratospheric ozone, [...]" The sentence is unclear and needs to be rewritten and probably split. Do the authors intend to say that the highest observed values of ozone at Summit are due to intrusion of stratospheric air masses in the course of tropopause folding events. This is correctly reproduced by the model?

**Authors' Response:**
We like to thank the reviewer for this suggestion, Line 353-355 is now split up to read: "At Summit, the time with the highest ozone level which occurs on April 18 is found by the model. The high ozone mixing ratio in the model is due to stratospheric ozone, reaching the troposphere due to a tropopause fold event as shown in Fig. 6 (unchanged in the revised paper)." We clarified that the large modeled ozone mixing ratio is clearly due to a tropopause folding event. Thus, we think it is highly likely that the large observed ozone mixing ratios are due to the same reason.

**Reviewer:**
L374–375: "[...] very small-scale structures such as open leads [...]" Open leads could also lead to local emissions of brominated VSLS from the ocean.

**Authors' Response:**
It is possible brominated VSLS are emitted from open leads, but we disregard it in the current model setup, which is specified on line 214 of the revised paper.

**Reviewer:**
L375–376: "[...] an accurate modeling of surface inversions might require very high vertical resolutions which are difficult to obtain in a synoptic scale simulation." Could you achieve this by nesting?

**Authors' Response:**
Vertical nesting is possible in WRF and may be combined with horizontal nesting, however, we think the current vertical resolution of the model is appropriate. As can be seen in Figure 7 and 8 of the revised paper, potential temperature profiles are predicted quite well by the model. The grid of WRF-Chem is non-equidistant in the vertical direction, see Table S2 of the supplement, which is similar to vertical nesting and allows to use a 25 m resolution in the lowest grid cells to resolve surface inversions.

**Reviewer:**
L406–409: "Some of the differences might be explained by a higher model resolution [...] resulting in more detailed structures in the model. Other differences [...] errors in the meteorology [...]." How does the uncertainty from satellite retrieval compare to the magnitude of divergence with modeling results?

**Authors' Response:**
The uncertainties of the satellite data contribute to the differences between model and observations. According to Sihler et al. (2012), they are typically below 50%. Accordingly, differences in absolute values between model and satellite measurement might be to a substantial part be caused by measurement uncertainties. However, the spatial patterns found in the satellite data are hardly affected because measurements, which are strongly influenced by clouds (cloud shielding), are filtered out using the sensitivity filter of 0.5 for the air mass factor of the lowest 500 m (Sihler et al., 2012). This discussion is added to the revised paper on lines 496-500.

**Reviewer:**
L503–505: "In a follow-up study it is planned to simulate ODEs in the year 2019 for which the new TROPOMI BrO VCDs with a high resolution of 5.5 km x 3.5 km are available. For this purpose, the grid resolution will be increased in order to allow for a comparison of the more refined observation data." You should connect this statement with the earlier introduction of

the OASIS dataset.

**Authors' Response:**
We incorporated OASIS results into the revised paper, which is also mentioned in the conclusions.

**Technical corrections**

**Reviewer:**
L58:"A further Br2 release mechanism [...] was suggested, this was also found in a laboratory study [...]" It would be better to split this sentence at the comma: "A further Br2 release mechanism [...] was suggested. Evidence for this mechanism was found in a laboratory study [...]."

**Authors' Response:**
We corrected the sentence as suggested.

**Reviewer:**
L80: "Temperatures [...] are likely to favour the occurrence of ODES [...]" typo "ODES"→"ODEs"

**Authors' Response:**
We corrected the typo.

**Reviewer:**
L104: "[...] 3D air quality model GEM-AQ. [...] the EMAC model [...]" These acronyms may need more explanation. For clarification: What are the major differences/improvements by using WRF-chem compared to the above mentioned models/simulations? GEM-AQ is much like WRF-Chem a weather prediction model with chemistry and aerosols able to run on different scales, while EMAC is a global chemistry climate model with a focus on middle atmosphere chemistry and dynamics.

**Authors' Response:**
We now explain the abbreviations in the text. In a regional model, it is easier to employ a finer grid size in both horizontal and vertical directions using WRF-Chem, which seems to be more difficult in the global models EMAC and GEM-AQ. For example, in order do study ODEs in the arctic, Toyota et al. (2011) employed a horizontal resolution of $0.9° \times 0.9°$ (approximately 100 km $\times$ 100 km) and 28 vertical grid cells with GEM-AQ, whereas Falk et al. (2018) used a horizontal resolution of $2.8° \times 2.8°$ (approximately 300 km $\times$ 300 km), both of which are much coarser than our 20 km $\times$ 20 km grid resolution. Also, WRF allows us to use the polar stereographic projection, which is more suitable for the arctic than a lat/lon grid. Currently, the boundary conditions strongly affect the results, but this can be cheaply resolved with an outer nest in a future investigation.

**Reviewer:**
L232: "[...] above a chosen sensitivity threshold of 0.5 are used.". Units?

**Authors' Response:**
The sensitivity threshold is applied for the air mass factor (AMF) for the lowest 500 m. This quantity has no unit, please see Sihler et al. (2012) for a detailed explanation.

**Reviewer:**
L425: typo "weighs"→"weights"

**Authors' Response:**
We corrected the typo.

**Reviewer:**
Fig. 6: Rainbow color color maps are generally depreciated for various reasons. First, it implies a distinct visual divergence of data at the edge between blue and green/yellow. This may lead to unintended misinterpretation () in some cases. Second, it is not colorblindness friendly. As the figure is purely used for illustration, chances of misinterpretations are low. Therefore, the authors may consider changing the color map, but it is not a must scientifically.

**Authors' Response:**
We used a rainbow color map for this particular plot in order to illustrate the location of the tropopause fold. In a perceptually uniform color map such as Viridis, which was used in the other 2D plots, the tropopause fold is less distinct.

**Referee #2**

**Reviewer Summary:**

**Reviewer:**
The modeling focuses on Feb. – Apr. 2009 with two types of comparisons to observational data: 1) Arctic-wide maps of BrO vertical column densities, and 2) ground-level O3 at Utqiaġvik, AK and Summit, Greenland. The time period was chosen to coincide with the OASIS field campaign at Utqiaġvik, AK during which a large suite of ground-level measurements were made "for comparison with the numerical results" (as the authors state on Lines 107-109), but no comparisons are made, even to the authors' own near-surface BrO data at Utqiaġvik, AK (Frieß et al., 2011, JGR). Comparisons to other available observational data, especially the available $Br_2$, HOBr, and BrO mole ratios, from the campaign would yield improved understanding of the simulations. For example, Figures 4 and 7 currently compare $O_3$ between two simulations and observations, but only modeled BrO is shown. Since BrO was measured at the same location and time, this lack of comparison is a major weakness of the current paper that was pointed out in the quick review. The authors stated in their response: "It is planned to include the OASIS observations of Frieß et al. (2001) and Liao et al. (2012), but this will be done at the next review stage of the paper... We will also investigate further OASIS data, which may be compared to the modeling results." I consider this to be an essential and critical step for model evaluation. Not only will this make the manuscript much stronger, but it is a huge opportunity, especially given the suite of available data, including the vertically resolved BrO profiles (for the dates and location in Figure 7!) in Frieß et al. However, since this has not yet been completed, I cannot currently evaluate this.

**Authors' Response:**
We again want to thank the reviewer for this suggestion. Figures 4 and 7 of the revised paper now include observed BrO by Liao et al. (2012) and the BrO profiles of Frieß et al. (2011), respectively. The figures are discussed in the corresponding text. See also our response to the fourth specific comment of Referee #1.

**Reviewer:**
The model is initiated with 0.3 ppt HBr and $Br_2$ in the lowest 200 m, and $CHBr_3$ is fixed to 3.5 ppt. Then $Br_2$ emissions occur via $BrONO_2$ and HOBr recycling on aerosols (for which the authors are encouraged to cite McConnell et al 1992, Nature (cited elsewhere in this manuscript) & Peterson et al. 2017, ACP, "Observations of bromine monoxide transport in the Arctic sustained on aerosol particles" for observational support of this mechanism). The authors then include snow $Br_2$ emission over sea ice via reaction of O3 with bromide and then claim "The bromine emission due to bromide oxidation by ozone is found to be important to provide an initial seed for the bromine explosion." (Abstract, Lines 10-12; also similarly stated on Lines 462-463 and 489-490) However, this is simply because of how the model is set up with this as the initial source, and it is not clear whether this setup and conclusion agrees with observations. Pratt et al. (2013, Nature Geoscience; which includes coauthor U. Platt) did not observe detectable $Br_2$ production upon ozone reaction with snow bromide in the dark, whereas $Br_2$ was produced when authentic snow was exposed to sunlight (without $O_3$), showing efficient condensed phase formation of $Br_2$. Addition of $O_3$ initiates the traditional bromine explosion mechanism and further $Br_2$ production (which is included in the model). The authors do not include condensed-phase $Br_2$ production from the snowpack and do state this in the manuscript as something for future work, but it is important when making conclusions statements that the model set-up and uncertainties be very clear, as this is otherwise misleading. There is an opportunity presented by this work in Figures 12, 13, and associated discussion to compare the modeled snow $Br_2$ emission rates to those previously measured by Custard et al. (2017, ACS Earth & Space Chem.) during Feb at Utqiaġvik and modeled by Wang & Pratt (2017, JGR).

**Authors' Response:**
We like to thank the reviewer for these comments. In the revised version of the manuscript, McConnell et al. (1992) and Peterson et al. (2017) are cited in section 2.3 (aerosol chemistry) for the recycling of bromine on aerosols, see line 175. We changed the dimensions of Figs. R7 and R8 (Figs. 12 and 13 of the revised paper) to molec $cm^{-2}$ $s^{-1}$, in order to allow an easier comparison with the literature. In the text referring to these figures, we added a discussion of $Br_2$ emissions in comparison to existing measurements and modeling studies: 'Emission rates of $Br_2$ from other studies are as follows: In February 2014, Custard et al. (2017) measured $Br_2$ fluxes of 0.07–1.2×$10^9$ molec $cm^{-2}$ $s^{-1}$ above the snow surface near Utqiaġvik with a maximum around noon. In a modeling study, Wang and Pratt (2017) found snowpack $Br_2$ emissions of 2.1×$10^8$ molec $cm^{-2}$ $s^{-1}$ on March 15, 2012 and 3.5×$10^6$ molec $cm^{-2}$ $s^{-1}$ on March 24, 2012. Emission fluxes due to the bromine explosion (HOBr + $BrONO_2$) are typically between 2-3×$10^9$ molec $cm^{-2}$ $s^{-1}$ (simulation 2) or 4-5×$10^9$ molec $cm^{-2}$ $s^{-1}$ (simulation 3) around noon and thus are on the higher end of the mentioned values. Bromine oxidation due to ozone, which plays the role of direct snowpack emissions in the present model, are rarely larger than 1×$10^9$ molec $cm^{-2}$ $s^{-1}$ with an average of around 2×$10^8$ molec $cm^{-2}$ $s^{-1}$ near Utqiaġvik, which compares quite well to the range found by Custard et al. (2017), while being larger than the values calculated by Wang and Pratt (2017).'

[Figure]

Figure R7: Emission rate of $Br_2$ due to $HOBr+BrONO_2$ (left) and due to bromide oxidation by ozone (center) from the snow surface for simulation 3, averaged over the complete simulation period. Ratio of $Br_2$ emissions due to HOBr and $BrONO_2$ to total $Br_2$ emissions on FY ice (right).

[Figure]

Figure R8: Emission rate of $Br_2$ due to HOBr and $BrONO_2$ and due to bromide oxidation by ozone from the snow surface at coordinates 178 W, 78 N for simulations 2 with $\beta = 1$ (left) and simulation 3 with $\beta = 1.5$ (right).

**Reviewer:**

In several locations in the Results & Discussion and Conclusions, the authors make statements that are not supported by the literature and modeling presented, especially when considering how the model is setup. Further the lack of comparison to BrO measurements at Utqiaġvik weakens the results presented because the ground-level halogen chemistry at ground level cannot be properly evaluated. It is my hope that the authors will add OASIS measurements of $Br_2$, BrO, and HOBr so that the model results can be better evaluated. Discussion that needs to be re-evaluated, in particular, includes: Lines 287-289 and Lines 342-343: "A possible conclusion is that the bromine explosion mechanism is inefficient to explain ODEs in the Arctic, or the present bromine explosion scheme is incomplete for instance with respect to emissions of bromide containing aerosols due to blowing snow and/or regions of increased B such as frost flowers." "...the bromine explosion mechanism alone is insufficient to properly predict the bromine production." I'm guessing that the authors may be referring here to the $Br_2$ production via $O_3$ reaction with $Br^-$ (R10) as the "bromine explosion mechanism", but this would not be consistent with the literature, that refers to the bromine explosion as R1 + R5 + R6 + R8 +R9.

**Authors' Response:**

As stated above in the response to reviewer #1, OASIS results are added to the sections 3.1 and 3.2 of the revised paper. With our statement "...the bromine explosion mechanism alone is insufficient to properly predict the bromine production.", we refer to simulation 4 in comparison to simulations 2 and 3. Simulation 4 strongly under-predicted the occurrence of ODEs. For simulation 4, $Br_2$ production via the $O_3$ reaction with $Br^-$ (R10) under sunlight was turned off, so that the emission probability for this reaction under sunlight is reduced from 7.5% to 0.1%. (R10) is thus nearly completely turned off for simulation 4, which is why we state that the bromine explosion, R1 + R5 + R6 + R8 + R9, is insufficient to explain the observed ODEs. We clarified this in the revised paper, starting on line 296: 'Quite a few ODEs are not captured by simulation 4, for which the emission probability for bromine emissions due to ozone under sunlight are reduced from 7.5% to 0.1%. Thus, direct emissions of bromine due to ozone are nearly completely turned off in simulation 4. This suggests a strong underestimation of bromine emissions without a direct emission of bromine due to ozone. A possible conclusion is that the bromine explosion mechanism is insufficient to explain ODEs in the Arctic [...]'

**Reviewer:**

Further, Wang et al. (2019, PNAS) showed, through simultaneous measurements of $Br_2$, HOBr, BrO, Br atoms, and $O_3$, that local ODEs could be quantitatively explained by the measured [Br], which would quantitatively show to be produced primarily by snow-phase reactions (condensed phase and heterogeneous). Further, the authors' statement is not in line with published literature and is further weakened by the fact that the modeling in the current work is missing what has experimentally been shown as a major source of primary $Br_2$. It needs to be acknowledged here that the model does not include sunlit condensed phase snowpack $Br_2$ production (without $O_3$), which Pratt et al. (2013, Nat. Geosc.) found to be an efficient $Br_2$ production mechanism, when $Br_2$ was not detectable from dark reaction of $O_3$ with snow, seaice, or brine icicles (proxy for frost flowers). Sunlit condensed-phase $Br_2$ production was also not observed for either sea ice or brine icicles, as predicted by Kalnajs and Avallone (2006, GRL), who stated "frost flowers are unlikely to be a direct source of atmospheric bromine."

**Authors' Response:**

In response to the reviewer's comment, we added a sentence at the end of section 2.4 (description of the emission scheme): 'Sunlit condensed-phase $Br_2$ production without any depositions of gas-phase species (Pratt et al., 2013, Wang et al., 2019) is currently not considered in the model.' We are uncertain, however, whether including a sunlit condensed phase snowpack $Br_2$ production would improve the model results. Wang and Pratt (2017) used emission rates $F \propto j_{Br_2}$, where the photolysis frequency of $Br_2$ is $j_{Br_2}$, or $F \propto j_{Br_2} [Br_2]$. With $F \propto j_{Br_2} [Br_2]$, sunlit condensed phase snowpack $Br_2$ production is not a starting mechanism for ODEs, so that it is no alternative to the bromide oxidation by ozone. With $F \propto j_{Br_2}$, no gas-phase bromine is needed for the emission to occur, so that sunlit condensed phase snowpack $Br_2$ production would be a starting mechanism of the bromine explosions. However, we believe this form of emission would produce a steady background concentration of bromine, since it only requires sunlight to occur, and would contribute to an over-prediction of BrO over FY ice. Additionally, there is a large uncertainty in the rate coefficient for this emission mechanism. For example, Wang and Pratt (2017) found for two days (March 15 and March 24) snowpack emission rates that differ by two orders of magnitudes. The current implementation of the emission mechanism is based on the deposition velocities and concentrations of gas-phase species such as ozone, which has been validated in previous studies, for instance, by Lehrer et al. (2004), Toyota et al. (2011), and Falk and Sinnhuber (2018).

**Reviewer:**

Lines 300-302: The authors discuss here the transport of BrO from the Bering Sea to Utqiaġvik, but it is important to remember that heterogeneous reactions are required in this scenario given the short lifetime of BrO (e.g., Platt and Honninger, 2003, Chemosphere; McConnell et al., 1992, Nature). Given their previous work, the authors are aware of this, but it needs to be clarified in the main text. Further, it is implied here that the model does not produce $Br_2$ near Utqiaġvik in February, but measurements at Utqiaġvik in February 2017 showed local measurable $Br_2$ snowpack fluxes (Custard etal 2017, ACS Earth & Space Chem). Further, U. Platt is a coauthor on the currently uncited, but highly relevant paper describing Jan.-Feb. BrO at Utqiaġvik (Simpson et al., 2018, GRL).

**Authors' Response:**

We clarified line 317: 'Since BrO over land is removed too quickly in the model, BrO can only be sustained through heterogeneous reactions while being transported from Bering Sea to Utqiaġvik by trajectories that go mostly over the sea ice.'

With the current model assumptions (no bromide and chloride content of the snowpack over land), $Br_2$ can indeed not be produced near Utqiaġvik, only recycled. Relaxing this assumption is not straightforward, however. We currently do not consider the snowpack in the model, which makes it difficult to prescribe a finite bromide and chloride content. Prescribing emission rates, as mentioned before, is also not straightforward due to the large range of reported snowpack emission rates. The paper of Simpson et al. (2018) is now cited in the discussion of BrO at Utqiaġvik, line 380.

**Reviewer:**

Lines 315-316: The authors discuss here model inaccuracies associated with elevated $O_3$, attributing this to haze or pollution. The prior modeling by Walker et al. (2012, JGR, "Impacts of midlatitude precursor emissions and local photochemistry on ozone abundances in the Arctic") is relevant here and should be considered in interpreting the authors' model results (including the separate statement on lines 355-356). Also, the model here does not spatially resolve leads, the convection from which have been experimentally shown to control $O_3$ recovery (end of the ODE) due to down-mixing from the free troposphere (Moore et al. 2014, Nature, "Convective forcing of mercury and ozone in the Arctic boundary layer induced by leads in sea ice"). The lack of spatial resolution of leads and resulting impacts on meteorology (only briefly mentioned on line 375) and ozone should be considered in the discussion and interpretation of model results

**Authors' Response:**

We added the following text to the discussion of arctic haze (line 336): 'Walker (2012) found that the decomposition of PAN, transported from lower latitudes or the upper troposphere to the arctic boundary layer, can account for up to 93% of the ozone production in the arctic.' For the discussion of open leads, line 398, we now write: 'Even that, however, might not be sufficient, since PBLs in the Arctic can be influenced by very small-scale structures such as open leads, which were found to play an important role in the ozone recovery after an ODE due to down-mixing of ozone-rich air from the free troposphere and which would require high-resolution sea ice data.' It should be noted however, that in Barrow (see Fig. 4 of the revised paper), we cannot find a mismatch of observed and modeled ozone that can be attributed to a slow recovery of ozone in the model, with the possible exception of March 16-17. We thus do not think, that the lack of spatial resolution for resolving open leads is a significant problem of the model.

**Reviewer:**

Lines 410-425 and Lines 483-488: This is good discussion, but it would further strengthen the discussion to incorporate observational literature to supports the interpretation here. For example, the model assumption that snow over land and near coasts have no salt content is inaccurate, as shown by Simpson et al. (2005, GRL) and supported by BrO measurements by Peterson et al. (2018, ACS Earth & Space Chem) and Pratt et al. (2013, Nature Geosci.), which are cited elsewhere in the manuscript. Similarly, the discussion of FY sea ice vs MY sea ice as a source of bromine is presented, with measured snow [Br$^-$] by Peterson et al. (2019, Elementa) and includes a section of suggestions to modelers.

**Authors' Response:**

We like to thank the reviewer for this comment and added additional discussion to line 453 of the revised paper, which clarifies that the assumption of no salt content of snow over land and near coasts is an idealization and may be improved in future studies: 'The assumption of zero bromide content of snow covering land or MY ice is of course an idealization and not always correct in reality (Simpson et al., 2005, Pratt et al., 2013, Peterson et al., 2018, Peterson et al., 2019), contributing to the under-prediction of BrO over land mentioned in this paragraph. Future simulations should aim to find ways to incorporate the salinity, pH, and the Br$^-$/Cl$^-$ ratio of the snowpack, which where found to be important parameters for the production of Br$_2$ (Pratt et al., 2013, Peterson et al., 2018, Peterson et al., 2019).' We want to stress, however, that the assumption of no bromide content of snow on land or near coasts can be correct in many circumstances, since bromide has to be supplied to the snow by e.g. bromide-rich aerosols being transported to the snow, which occurs more often during storms.

**Additional comments**

**Reviewer:**

Lines 8-10: I suggest leaving the statements about the reactive surface ratios out of the abstract, as there isn't sufficient context here for the reader to understand what this physically means. It would be better to instead discuss other scientific results of the work, especially since this surface ratio seems to effectively be a model tuning parameter.

**Authors' Response:**

We now avoid using the term 'reactive surface ratio' in the abstract, instead we mention faster emissions on lines 10-11. The comparison with OASIS data is mentioned in the abstract of the revised paper. Two sentences in the abstract discuss new findings: 'Bromine release due to N$_2$O$_5$ was found to be important from February to mid March, but irrelevant thereafter. A comparison of modeled BrO with in-situ and MAX-DOAS data hints at missing bromine release and recycling mechanisms on land or near coasts.'

**Reviewer:**

Lines 22-23: Note that Br atom reaction with ozone was directly observed by Wang et al. (2019, PNAS), so this can be stated more strongly here by inclusion of this reference.

**Authors' Response:**

We added the suggested reference to line 22.

**Reviewer:**

Lines 32-33: Replace "ice" with "snow" to reflect current knowledge based on more recent direct

measurements: Pratt et al.. (2013, Nature Geoscience), Custard et al. (2017, ACS Earth & Space Chem.). Further, Pratt et al. (2013, Nature Geoscience) showed through direct measurements that sea ice did not produce detectable $Br_2$. Further, while there is still much to be learned about $Br_2$ production, as reflected in the sentence on lines 33-34, this paragraph primarily cites references from the 1990s and does not discuss current knowledge gained from the last decade, thereby suggesting that less is known.

**Authors' Response:**
We added the references to Pratt et al. (2013) and Custard et al. (2017) to lines 35-36 and changed ice to snow.

**Reviewer:**
Line 40: The authors chose to designate heterogeneous reactions with "aq" above the arrow, but this is not a common notation. It would be clearer in the introduction to simply include the phases of each species as subscripts next to them in the equations so that it is clear to a novice reader which species are in the gas vs aqueous phase.

**Authors' Response:**
Although this kind of notation is found in the literature, we agree with the reviewer that it is more clear to include the phases of each species for heterogeneous reactions instead of writing an "aq" above the arrow. For example,

$$HOBr + H^+ + Br^- \xrightarrow{\text{aq}} Br_2 + H_2O$$

is modified to read

$$HOBr\,(g) + H^+(aq) + Br^-(aq) \longrightarrow Br_2(g) + H_2O\,(l)$$

in the revised manuscript.

**Reviewer:**
Line 45: The authors can strengthen this sentence by pointing to Pratt et al. (2013, Nature Geoscience), who directly showed the pH dependence of $Br_2$ production through field-based experiments

**Authors' Response:**
We added the sentence "A pH-dependence of the $Br_2$ production was shown by Pratt et al. (2013) through field-based experiments." to the line.

**Reviewer:**
Line 57: Please clarify what is meant by "the surface layer" here. I believe the authors are discusses the snow grain surface, but this phrasing is also used for the atmosphere. Also, the authors should note, for clarity here, that sunlight is required for condensed-phase OH production

**Authors' Response:**
We clarified the sentence: "A further $Br_2$ release mechanism initiated by a reaction of the hydroxyl radical OH with bromide inside the surface layer of the snow grains under sunlight was suggested [...]"

**Reviewer:**

Lines 61-67: References are needed for these sentences. The work of Thompson et al. (2015, ACP, "Interactions of bromine, chlorine, and iodine photochemistry during ozone depletions in Barrow, Alaska") is highly relevant here, especially since that work focused on the OASIS field campaign.

**Authors' Response:**

We added the reference Thompson et al. (2015) to line 68.

**Reviewer:**

Lines 75-76: The authors cite a 2007 review here for BrCl production via HOBr + Cl$^-$. It is important to note that McNamara et al. (2020, ACS Earth & Space Chem), who measured BrCl, showed that the Cl2 + Br$^-$ reaction is also a significant source of BrCl in the Arctic spring. However, despite mentioning these sources of Br atoms, Tables S3 and 4 do not include these reactions, suggesting that they are not in the model. Please clarify this in the text, as it is an additional uncertainty associated with the model setup.

**Authors' Response:**

These reactions are not included in the model. This is now mentioned in line 173 of the revised paper.

**Reviewer:**

Lines 80-81: Note that more recent work by Halfacre et al. (2014, ACP) showed ozone measurements from five buoys across the Arctic and "no apparent dependence [of ODEs] on local temperature".

**Authors' Response:**

We now end line 81 with '[...] and Halfacre et al. (2014), using ozone measurements at five buouys across the Arctic, found no apparent temperature dependence for the presence of an ODE.'

**Reviewer:**

Line 89: Nasse et al. 2019 (Atmos Meas Tech) appears to be the incorrect reference here, as it does not include Arctic ozone data.

**Authors' Response:**

Thank you, we meant to cite the dissertation of Jan-Marcus Nasse in this line (`https://archiv.ub.uni-heidelberg.de/volltextserver/26489/`) and instead we erroneously cited a technical paper with the same main author. This is corrected in the revised paper on line 89. It should be noted, that the dissertation only includes partial ODEs in the Antarctic. This is also clarified in the revised text: 'ODEs are much less pronounced in polar fall with rare measurements of partial ODEs in the Antarctic (Nasse 2019), [...]' We are not aware of any ODEs observed in the arctic fall, although it is likely that they also exist.

**Reviewer:**

Lines 93-94: Please connect this sentence about Thomas et al. (2011) to the halogen chemistry being discussed in this paragraph.

**Authors' Response:**
We added a sentence to line 101: 'They found the solar actinic flux to be the main driver of reactive bromine release from the liquid-like layer (LLL) of the snow grain surface and a dependence of bromine release from the LLL on the OH concentration in the LLL.'

**Reviewer:**
Figure 1: Provide the source of the sea ice data. Also label the latitude rings and especially where 88N is, and state in the caption that FY ice is assumed above this latitude, as stated in the text.

**Authors' Response:**
We added a citation of the source of the sea ice data (also cited in Table 1) to the caption of Fig. 1 of the revised paper and added the sentence 'For latitudes larger than 88°, missing sea ice type data is filled up with FY ice.' to the caption. Additionally, the latitude rings are labeled with an additional ring at 88N.

**Reviewer:**
Lines 141-142: Note that iodine atom reaction with ozone has been shown to significantly increase the ozone depletion rate, even at low mole ratios (e.g. Raso et al. 2017, PNAS, showed 0.3 ppt of $I_2$ to increase the initial rate of ozone depletion by 31% in a case study). Adding iodine chemistry to the model would be a significant task, so I think it is ok currently to not include. However, a greater acknowledgement of uncertainty is required here and should be mentioned again in the discussion and conclusions when comparing ozone levels between measurements and modeling.

**Authors' Response:**
We acknowledge the uncertainties involved in neglecting iodine on lines 156-159 of the revised paper: 'Observations of reactive iodine in the arctic region (Zielcke et al., 2015, Raso et al., 2017) suggest only low mixing ratios of iodine. Already small mixing ratios of iodine can significantly enhance ozone depletion (Raso et al., 2017), however, iodine is still neglected due to the uncertainties in the abundance of iodine in the arctic atmosphere and snowpack.' In line 510 of the conclusions, these uncertainties are also acknowledged: 'Iodine chemistry was neglected in this study, which may contribute to the under-prediction of ODEs at Utqiaġvik.'

**Reviewer:**
Line 170 and Table S3 caption: Change "ice/snow" to "snow" here, since Line 191 states that all ice is assumed to be snow-covered.

**Authors' Response:**
We changed ice/snow to snow in L170 and the caption of S3, as suggested.

**Reviewer:**
Line 193: This is the only place that I saw $N_2O_5$ deposition mentioned as a source of Br2. This is intriguing and should be discussed in the results. What level of $N_2O_5$ is predicted by the model, and what fraction of the $Br_2$ is predicted to be produced? I'm also confused because I don't see this heterogeneous reaction in Table S3. McNamara et al. (2019, Environ. Sci. Technol.) measured $N_2O_5$ at Utqiaġvik during spring, which would provide a comparison point.

**Authors' Response:**

The reviewer might have overlooked that the $N_2O_5$ emission had already been mentioned in Tab. S3 of the supplement of the discussion paper:

$$N_2O_5(g) + Br^-(aq) + H^+(aq) \longrightarrow BrNO_2(g) + HNO_3(g).$$

$BrNO_2$ is photolyzed under sunlight into Br and $NO_2$. The mixing ratio of $N_2O_5$ in the lowest grid cell averaged over one month is shown in Fig. R9 (Fig. S5 in revised supplement) for February, March, and April 2009. Large concentrations of $N_2O_5$ mostly occur in February. Near North Alaska, $N_2O_5$ is produced by anthropogenic emissions at Prudhoe Bay as predicted by the EDGAR-HTAP anthropogenic emission dataset. $N_2O_5$ over the Arctic Archipelago is produced by anthropogenic emissions over Baffin Island. $N_2O_5$ over Siberia is partially produced by anthropogenic emissions resolved by the model, but mostly advected from the boundary conditions. In February, $N_2O_5$ is very stable due to a lack of sunlight and low temperatures, which explains the large concentrations and decrease in the following months. Since most of the bromine is produced in March and April over FY ice, bromine production due to $N_2O_5$ has little relevance for the second half of the simulation, but is relevant for the end of February and early March. Bromine emissions due to $N_2O_5$ in February are unlikely to start a bromine

[Figure]

Figure R9: Mixing ratio of $N_2O_5$ in the lowest grid cell, averaged over one month for February (left), March (center) and April (right).

[Figure]

Figure R10: Mixing ratio of ozone, $Br_2$ and $N_2O_5$ at Utqiaġvik for simulation 3.

explosion due to the lack of sunlight in the northern regions. However, Namara et al. (2019) measured $N_2O_5$ near Utqiaġvik and found 50 pmol $mol^{-1}$ of $N_2O_5$ for one day in mid March and between 0-15 pmol $mol^{-1}$ otherwise from March to April. Figure R10 (Figure S6 in the revised supplement) shows modeled $N_2O_5$ in March and April at Utqiaġvik. Most of the time, the $N_2O_5$ mixing ratio is between 0 and 15 pmol $mol^{-1}$, but at a few days, where more than 50 pmol $mol^{-1}$ of $N_2O_5$ is predicted. On March 8, a mixing ratio of approximately 300 pmol $mol^{-1}$ is found by the model. All of the events with enhanced mixing ratios of $N_2O_5$ are caused by advection of polluted air from Prudhoe Bay to Utqiaġvik. This discussion is added to the supplement, section F, together with the relevant figures.

**Reviewer:**
Lines 234-245: It is suggested to move this text and Table 2 to the Methods section, as results are not presented here.

**Authors' Response:**
We moved the text and Table 2 to the Methods section, which is the new subsection 2.6 of the revised Paper.

**Reviewer:**
Line 281 and Figure 4 caption: It is stated that "Figure 4 shows modeled and observed surface ozone and BrO at Utqiaġvik" when measured BrO is not shown, and this is misleading. It is my hope that the authors will rectify this by adding measured BrO (from the OASIS study) to Figure 4.

**Authors' Response:**
BrO from the Oasis campaign is included in the new Fig. 7 of the revised paper.

**Reviewer:**
Lines 286, 341-343, and Lines 472-473: "This suggests a strong underestimation of BrO emission without a direct emission of BrO due to ozone." "...a reduced BrO emission due to direct bromide oxidation by ozone..." "BrO may be emitted by the extended bromine explosion mechanism and/or oxidation of bromide by ozone directly from the sea ice." These statements need to be rephrased as BrO is not directly emitted from snow or sea ice.

**Authors' Response:**
We changed BrO in these sentences to bromine.

**Reviewer:**
Table 4: Why is simulation 2 shown here for Summit, instead of simulation 3, which was deemed the best setup for Utqiaġvik (since simulation 2 missed half of the ODEs at Utqiaġvik)? This should be discussed.

**Authors' Response:**
We agree with the reviewer, it makes more sense to use simulation 3 as we deemed it to be the best simulation. Simulation 3 is now shown instead of simulation 2 in Table 4 at Summit. We also added simulation 1 (no halogen chemistry) as a reference to the table. There is little difference between the simulations at Summit, since no ODEs occurred there in the modeled time period.

**Reviewer:**
Line 350: Fig. 5 doesn't show Summit results as stated here.

**Authors' Response:**
We used the wrong reference, Fig. 4 of the paper was meant, thank you! This was corrected in the revised manuscript.

**Reviewer:**
Figures 9-12, 14: Please make the latitude and land lines more visible to aid interpretation of the plots. Also, clarify in the Figure 9 caption what is meant by "cone".

**Authors' Response:**
The latitude and land lines are thicker and thus more visible in the revised paper. We now write 'segment' or 'segment of a circle' instead of 'cone' in both the caption of Fig. 9 and the text of the paper. A 60° segment of a circle, centered at the north pole, is assigned to each time point.

**Reviewer:**
Figure 12: Does this include only snow, or snow + aerosols? Please clarify.

**Authors' Response:**
This only includes emissions from the snow surface. It is now clarified in both the figure caption and the text.

[revised manuscript text omitted]

---

## Author Response (AR2)

**Authors' response to the reviewer**

The authors like to thank the reviewer again for the valuable comments. We revised the manuscript where modifications of the text are marked in blue color. In particular, we address the comments in detail as follows.

**Reviewer:**

Herrmann et al revised their manuscript describing WRF-Chem simulation of Arctic ODEs and comparison to observations in Utqiagvik, Alaska and Summit, Greenland. The major revision that was completed was the addition of comparison of model results with ground-based observations at Utqiagvik during the OASIS campaign. This was a major undertaking, and the authors are commended for doing this, as this addition significantly strengthened the manuscript, which now adds to the great body of literature stemming from the OASIS campaign. The added comparisons to this data and associated published literature further supports the authors' results, elevating the impact of the results. My comments below focus on the newly added figures and text, with line numbers referring to the tracked changes version of the manuscript.

Figure 4: In response to Reviewer #2 (top of page 14), the authors state that Figure 4 now includes observed BrO (which would be excellent), but I do not see that in either the tracked changes or manuscript file. Perhaps the figure was accidentally not updated?

**Authors' Response:**

We erroneously wrote that observed BrO is included in Figure 4. We tried to add observed BrO to Figure 4 in an earlier version of the revised manuscript, however, it was extremely difficult to distinguish the lines. As a compromise, we decided to instead add Figure 7 to the paper, which contains only measured and observed BrO and measured $BrNO_2$ for the time range of the observed BrO. We improved Figure 7 of the revised paper, see Figure R1 of this answer. We swapped the colors of measured BrO and simulated $BrNO_2$ and added ozone of simulation 3, which makes a comparison to Figure 4 of the manuscript easier. Also, we changed the aspect ratio of the figure.

**Reviewer:**

The addition of Figures 7 and S5 and the associated new discussion of simulated production of $BrNO_2$ from reaction of $N_2O_5$ with $Br^-$ during oil field influence is really interesting and a

[Figure]

Figure R1: Comparison of modeled BrO and in-situ observations of BrO at Utqiaġvik (Liao et al., 2011) and modeled $O_3$ and $BrNO_2$; the numerical results are for simulation 3. The date shown is for 00:00, GMT+0.

[Figure]

Figure R2: Modeled $NO_x$, BrO, HOBr and $BrONO_2$ at Utqiaġvik for simulation 3. The date shown is for 00:00, GMT-9 (local time at Utqiaġvik, GMT-9). The time range and timezone is chosen to be directly comparable to Figure 7 of Custard et al., 2015.

great addition to the paper. In particular, higher $BrNO_2$ is predicted earlier during the OASIS campaign when NOx was elevated. This discussion would benefit significantly from integration of discussion of the results of Custard et al (2015, Atmos. Chem. Phys., "The NOx dependence of bromine chemistry in the Arctic atmospheric boundary layer"), who completed 0-D modeling of OASIS, examining the role of NOx in bromine chemistry and predicting BrNO2 production during the same time period as simulated in the current work.

**Authors' Response:**

Thank you for the suggestion. Figure R2 shows modeled $NO_x$, BrO, HOBr, and $BrONO_2$ at Utqiaġvik for simulation 3, which may be compared to the results of Custard et al. (2015). Figure R2 is added to the supplement, as a subfigure of S6. The following discussion is added to line 400 of the revised paper: 'Custard et al. (2015) studied the role of NOx in bromine chemistry from the March 24, 2009 to April 3, 2009 at Utqiaġvik using a box model. They found a suppression of ozone destruction for a high $NO_x$ case (concentrations in the range of 800 to 1600 pmol/mol). During this time frame, the simulation with WRF-Chem predicts negligible production of reactive bromine due to $N_2O_5$. In Fig. S6 of the supplement, modeled Modeled $NO_x$, $BrONO_2$, HOBr and BrO is shown for the time range modeled in Custard et al. (2015). Modeled $NO_x$ is elevated from March 24 to March 26 and again on April 2, similar to the measurements of $NO_x$ shown in Fig. 2 of the paper of Custard et al. (2015). However, the present model does not find $NO_x$ mixing ratios on the order of 10,000 pmol/mol as found on March 24-27 in the measurements. The typical modeled NOx concentrations are in the range of 50 to 1000 pmol mol$^{-1}$, i.e. between the high and low $NO_x$ scenarios of Custard et al., 2015. The predicted values of $BrONO_2$ compare quite well with these of Custard et al. (2015), see Fig. 7c of that work, with peak values around 50 pmol/mol.'

**Reviewer:**

Several places in the manuscript refer to an under-prediction of BrO over land, and it is stated that this is discussed in a later section (presumably the brief mention on page 28?). To support this, I suggest adding a sentence on Line 509 that refers to Pratt et al (2013, Nat. Geo.) and Peterson et al. (2018, ACS Earth & Space Chem), both of which report MAX-DOAS BrO observations over the tundra snowpack (up to >100 km inland). In fact, in the Utqiagvik region, Peterson et al (2018) observed higher BrO over the tundra than the FYI. Adding a sentence referring to previous measurement of BrO over inland tundra snowpack provides an explanation for the current study's result (given the short lifetime of BrO) and will strengthen the manuscript as a result.

**Authors' Response:**

We added the following text to line 492 of the revised manuscript: 'Pratt et al. (2013) and Peterson et al. (2018) reported BrO observations using MAX-DOAS over the tundra snowpack, which show elevated BrO levels up to more than 100 km inland. Peterson et al. (2018) found higher BrO concentrations over the tundra than over FY ice. In contrast to that, the simulations conducted in this work under-predict BrO over land...'

**Additional comments**

**Reviewer:**
Line 2: In response to Reviewer #1, the authors changed "studied using the regional software WRF-Chem" to "studied using the open-source software package WRF-Chem". However, this didn't address the reviewer's comment. It is important for someone not familiar with WRF-Chem to understand that it is a regional model, and not another type of software.

**Authors' Response:**
Thank your for the suggestion, we agree that it is important to mention the usage of a regional model. In the revised paper, line 2 reads '...are studied using a regional model based on the open-source software package WRF-Chem.'

**Reviewer:**
Line 4: In response to Reviewer #1, the authors clarified elsewhere that Br2 is emitted from snow above sea ice rather than the sea ice itself, but this sentence still needs to be updated by deleting "ice and". Also, Lines 577-578 needs to be revised as well, as it states "...oxidation of bromide by ozone directly from the sea ice."

**Authors' Response:**
We removed the words 'ice and' from line 4. Lines 577-578 of the revised manuscript now states 'Bromine may be emitted by the extended bromine explosion mechanism and/or oxidation of bromide by ozone directly from the snow covering sea ice.'

**Reviewer:**
Line 25: I suggest changing "is most likely destroyed" to "is destroyed".

**Authors' Response:**
We changed line 25 as suggested.

**Reviewer:**
Line 49: I suggest adding the following to this new sentence "...field-based experiments, and Wren et al. (2013, ACP) and Halfacre et al. (2019, ACP) through lab-based experiments."

**Authors' Response:**
We modified line 49 as suggested.

**Reviewer:**
Lines 214-215: I suggest replacing Wang et al. (2019b) here with the more appropriate Halfacre

et al. (2019, ACP).

**Authors' Response:**
We reworded the sentence slightly and changed the reference as suggested.

**Reviewer:**
Figure 7: The addition of this figure is quite valuable. However, it is very difficult to discern the black trace as currently plotted. Also, the time zone plotted needs to be stated, either in the figure or caption, in this figure and all other similar figures (Figures 2, 4, 10, 11, 16). I also suggest removing the "00:00" from the x axis labels and instead label more ticks to make it easier to interpret by having more dates labeled.

**Authors' Response:**
The time zone is now stated in the captions of the Figures in the revised manuscript. Also, we improved the Figures as suggested. Figure R1 shows the revised version of Figure 7 of the manuscript.

**Reviewer:**
Figures 6, 8, & 9: Please provide the time zone that the vertical profile times correspond to.

**Authors' Response:**
The time zone now is mentioned in the caption of Figures 6, 8, and 9.

**Reviewer:**
Figure 9 bottom-right: If the BrO observations above 100 m are known to be inaccurate as stated in the caption, then they should not be plotted in the figure.

**Authors' Response:**
We changed the figure as suggested, see Figure R3, which replaces Fig. 9 in the manuscript.

**Reviewer:**
Lines 441-442: Reference to Moore et al. (2014, Nature) should accompany the added text here.

**Authors' Response:**
We added the reference to Moore et al. (2014, Nature) to the suggested lines.

**Reviewer:**
Line 508: Rather than citing Pratt et al. (2013) here, I suggest adding Jacobi et al. (2012, JGR, "Chemical composition of the snowpack during the OASIS spring campaign 2009 at Barrow, Alaska) since it describes the snow composition during the OASIS campaign, and Krnavek et al. (2012, Atmos. Environ., "The chemical composition of surface snow in the Arctic: Examining marine, terrestrial, and atmospheric influences"), since it compares [Br-] over tundra, FYI, and MYI, to the publications already listed. I encourage the authors to review these and other Arctic snow composition measurements studies, which refute their statement on page 18 of the response that "the assumption of no bromide content of snow on land or near coasts can be correct in many circumstances", as this statement is not supported by measurements.

[Figure]

Figure R3: Vertical profiles of measured and modeled (simulation 2 (left) and simulation 3 (right)) ozone, of potential temperature θ, and of BrO at Utqiaġvik on March 22 (top) and April 15 (bottom), 2009. The time zone is GMT+0. Measurements are from upward flights using ozone sondes (Oltmans et al., 2012) and DOAS measurements (Frieß et al, 2012). On April 15, only the observed BrO mixing ratio in the lowest 100 m is accurate due to very poor visibility.

**Authors' Response:**

We added Jacobi et al. (2012) and Krnavek et al. (2012) to Line 508 of the revised manuscript, as suggested. We are aware of the fact that the bromide content of snow can be non-zero. However, with regard to bromine emissions from the snow, it is sufficient for the bromide content to be sufficiently small for the assumption of no bromide content to be approximately correct. In the manuscript we write: 'The assumption of zero bromide content of snow covering land or MY ice is of course an idealization and not always correct in reality (Simpson et al., 2005; Jacobi et al., 2012; Krnavek et al., 2012; Peterson et al., 2018, 2019), contributing to the underprediction of BrO over land mentioned in this paragraph.' We note that the measurements of snow bromide span a range of nearly three orders of magnitude (Jacobie et al., 2012, find 5 to 1400 $\mu$g/l, depending of the type of snow).

**Reviewer:**

Response Page 16: Note that Wang and Pratt (2017) simply used jBr2 as a term to define the timing of radiation-dependent emission of Br2 from the snowpack as a representation of the observations by Pratt et al. (2013), which showed the Br2 was produced from Arctic snow in a chamber only upon irradiation and no addition of O3 or other gas-phase oxidant. This mechanism was replicated in the lab by Halfacre et al. (2019, ACP), who showed Br2 production upon irradiation of ice containing Br- and an OH precursor. Therefore, it is incorrect that Br2 is required for condensed-phase snowpack Br2 production.

**Authors' Response:**

We are sorry for the misunderstanding. In Wang and Pratt (2017), section 3.2, two different parameterizations of snowpack emissions are described:
'(i) JScale: emission rates (FX2) are scaled linearly with j-values (i.e., $F_{X2} \propto j_{X2}$, where X2 = $Cl_2$ or $Br_2$);'
'(ii) SS: emission rates are scaled with steady-state (SS) removal (i.e., $F_{X2} \propto j_{X2} \cdot [X2]$)). The SS parameterization is based on the steady-state assumption that the snowpack emission rates of $Br_2$ and $Cl_2$ are balanced by their photolysis, due to the short daytime photolysis lifetimes of $Br_2$ and $Cl_2$ in the Arctic (tens of seconds and tens of minutes, respectively, for March 2009 in Utqiaġvik;'
We tried to address both parameterizations in our response, saying that parameterization (ii) should be, in our understanding, not suitable to describe the initiation of a bromine explosion, since the parameterization assumes the emission of reactive bromine to be proportional to the concentration of $Br_2$. Therefore, at zero Br it will produce a zero Br-flux and thus a Br-explosion can never start. Parameterization (i), which is addressed later in the same response on page 16, of course does not require $Br_2$, as the reviewer state, and thus can lead to the start of a Br-explosion. The manuscript was not changed in response to this question.

**Reviewer:**

Line 623-624: I encourage the authors to add acknowledgement of the individuals that conducted the OASIS measurements and produced the data used. These BrO measurements by CIMS and DOAS are not trivial whatsoever, and these individuals should at least be recognized here, as their data significantly contributed to the manuscript. While H. Sihler and U. Platt are listed under the author contributions, there were other individuals that conducted the CIMS measurements, in particular.

**Authors' Response:**

Thank you, in the revised manuscript, we thank J. L. Liao, L.G. Huey and D. J. Tanner, who conducted the CIMS measurements: 'The authors thank J. Liao, L. G. Huey and D. J. Tanner,

who conducted CIMS measurements during the OASIS campaign.'. U. Frieß and H. Sihler, who conducted the DOAS measurements, are co-authors of the present paper.

**References**

Jacobi, H. W., Voisin, D., Jaffrezo, J. L., Cozic, J., and Douglas, T. A.: Chemical composition of the snowpack during theOASIS spring campaign 2009 at Barrow, Alaska, Journal of Geophysical Research: Atmospheres, 117, 2012.

Krnavek, L., Simpson, W. R., Carlson, D., Domine, F., Douglas, T. A., and Sturm, M.: The chemical composition of surface snow in the Arctic: Examining marine, terrestrial, and atmospheric influences, Atmospheric Environment, 50, 349–359, https://www.sciencedirect.com/science/article/pii/S1352231011012192, 2012

Moore, C. W., Obrist, D., Steffen, A., Staebler, R. M., Douglas, T. A., Richter, A., and Nghiem, S. V.: Convective forcing of mercury andozone in the Arctic boundary layer induced by leads in sea ice, Nature, 506, 81, 2014
Peterson, P. K., Pöhler, D., Zielcke, J., General, S., Frieß, U., Platt, U., Simpson, W. R., Nghiem, S. V., Shepson, P. B., Stirm, B. H., and Pratt, K. A.: Springtime Bromine Activation over Coastal and Inland Arctic Snowpacks, ACS Earth and Space Chemistry, 2, 1075–1086, https://doi.org/10.1021/acsearthspacechem.8b00083, 2018.

Peterson, P. K., Hartwig, M., May, N. W., Schwartz, E., Rigor, I., Ermold, W., Steele, M., Morison, J. H., Nghiem, S. V., and Pratt, K. A.: Snowpack measurements suggest role for multi-year sea ice regions in Arctic atmospheric bromine and chlorine chemistry, Elementa 775 (Washington, DC), 7, 2019.

Pratt, K. A., Custard, K. D., Shepson, P. B., Douglas, T. A., Pöhler, D., General, S., Zielcke, J., Simpson,W. R., Platt, U., Tanner, D. J., et al.: Photochemical production of molecular bromine in Arctic surface snowpacks, Nature Geoscience, 6, 351, 2013.

Simpson, W. R., Alvarez-Aviles, L., Douglas, T. A., Sturm, M., and Domine, F.: Halogens in the coastal snow pack near Barrow, Alaska: Evidence for active bromine air-snow chemistry during springtime, Geophysical research letters, 32, 2005.

Wang, S. and Pratt, K. A.: Molecular Halogens Above the Arctic Snowpack: Emissions, Diurnal Variations, and Recycling Mechanisms, Journal of Geophysical Research: Atmospheres, 122, 11,991–12,007, https://agupubs.onlinelibrary.wiley.com/doi/abs/10.1002/2017JD027175, 2017.